# Human TRIM5α: Autophagy Connects Cell-Intrinsic HIV-1 Restriction and Innate Immune Sensor Functioning

**DOI:** 10.3390/v13020320

**Published:** 2021-02-19

**Authors:** Alexandra P. M. Cloherty, Anusca G. Rader, Brandon Compeer, Carla M. S. Ribeiro

**Affiliations:** Amsterdam UMC, University of Amsterdam, Department of Experimental Immunology, Amsterdam Institute for Infection & Immunity, Meibergdreef 9, 1105 AZ Amsterdam, The Netherlands; a.p.cloherty@amsterdamumc.nl (A.P.M.C.); a.g.rader@amsterdamumc.nl (A.G.R.); brandoncompeer@hotmail.com (B.C.)

**Keywords:** TRIM5α, autophagy, HIV-1 restriction, viral evasion, antiviral immunity, Langerhans cells, Langerin, dendritic cells, macrophages, CD4^+^ T cells

## Abstract

Human immunodeficiency virus-1 (HIV-1) persists as a global health concern, with an incidence rate of approximately 2 million, and estimated global prevalence of over 35 million. Combination antiretroviral treatment is highly effective, but HIV-1 patients that have been treated still suffer from chronic inflammation and residual viral replication. It is therefore paramount to identify therapeutically efficacious strategies to eradicate viral reservoirs and ultimately develop a cure for HIV-1. It has been long accepted that the restriction factor tripartite motif protein 5 isoform alpha (TRIM5α) restricts HIV-1 infection in a species-specific manner, with rhesus macaque TRIM5α strongly restricting HIV-1, and human TRIM5α having a minimal restriction capacity. However, several recent studies underscore human TRIM5α as a cell-dependent HIV-1 restriction factor. Here, we present an overview of the latest research on human TRIM5α and propose a novel conceptualization of TRIM5α as a restriction factor with a varied portfolio of antiviral functions, including mediating HIV-1 degradation through autophagy- and proteasome-mediated mechanisms, and acting as a viral sensor and effector of antiviral signaling. We have also expanded on the protective antiviral roles of autophagy and outline the therapeutic potential of autophagy modulation to intervene in chronic HIV-1 infection.

## 1. HIV-1 Infection: Decades after Implementation of Antiretroviral Therapy

Human immunodeficiency virus-1 (HIV-1), the causative pathogen of acquired immune deficiency syndrome (AIDS) in humans, persists as a global health concern with an incidence rate of 1.7 million in 2019 and an estimated global prevalence of 38 million [1]. To counteract the global HIV-1 epidemic, combination antiretroviral therapy (cART) has been developed, and has demonstrated high levels of efficacy, safety and tolerability for people living with HIV (PLWH) [2]. cART treatment is typically composed of multiple antiretroviral drugs that target diverse steps of the HIV-1 replication cycle, including nucleoside reverse transcriptase inhibitors, non-nucleoside reverse transcriptase inhibitors, protease inhibitors, integrase inhibitors, and fusion/entry inhibitors [3]. cART treatment has been highly effective in reducing viral burden, ultimately resulting in prolonged life expectancy of PLWH [2,4,5]. However, despite the effectiveness of cART treatment, it is not curative. cART-treated HIV-1 patients exhibit viral latency, i.e., the ability of the virus to reside dormant within a cell, in the form of integrated viral DNA within long-lived tissue resident cells, and increased risk for chronic inflammation [6,7,8,9,10,11,12]. Whether this chronic inflammation is due to the treatment, residual inflammation, or viral latency is unknown. HIV-1 persistence in reservoirs is the major obstacle to an HIV-1 cure, as viral replication rapidly resumes after cART interruption.

Major HIV-1 cellular reservoirs, within which HIV-1 latently persists, include differentiated memory and effector CD4^+^ T-cell subsets, as well as tissue-resident resting myeloid cells [6,7,8,13]. These cells are not targeted by cART since they do not actively transcribe the virus, thereby resulting in survival of latent HIV-1 in the form of integrated viral DNA and permitting formation and maintenance of HIV-1 reservoirs [9]. Therefore, PLWH are required to take cART for life, while remaining at a higher risk than age-matched controls for mortalities and morbidities such as dementias, malignancies, and cardiovascular disease [11]. In order to circumvent these cART limitations, and to ultimately achieve HIV-1 cure, alternative combinatory HIV-1 therapeutic strategies are imperative.

Although cART was originally developed as a treatment for PLWH, recently it has also been adapted for pre-exposure prophylaxis (PrEP) for high-risk individuals including men who have sex with men (MSM), sex workers, and injection drug users. While PrEP has been demonstrated to be a successful preventative therapy, there are specific populations for which PrEP shows reduced efficacy, as reviewed in [14]. For instance, amongst young women with a high risk of HIV-1 infection, PrEP was demonstrated to be less effective in those with genital inflammation as compared to those without genital inflammation [15,16]. Hence, it is important to develop strategies that better protect groups with high risk for contracting HIV-1, and particularly for groups in which existing PrEP strategies show reduced efficacy.

### 1.1. Innate Restriction Factors Are Natural Host Factors That Interfere with HIV-1 Replication

Many efforts have been made to develop new drugs or combinational therapeutics together with cART [17,18,19,20]. One promising approach is to capitalize on antiviral restriction factors, which are innate host proteins that interfere with viral replication and therefore serve as a natural defense against HIV-1. A particular advantage of harnessing restriction factors for use as a prophylactic or therapeutic strategy against HIV-1 is the capacity of restriction factors to not only directly restrict HIV-1, but also to enhance innate and adaptive immune responses against the virus [21]. In general, restriction factors are interferon (IFN)-inducible, and their recognition of viral components often triggers innate immune activation which subsequently interferes with the virus replication cycle [22,23,24,25,26].

Although restriction factors have several shared characteristics, they utilize distinctive mechanisms to block specific steps of the viral replication cycle [21,27]. The HIV-1 replication cycle can be divided into eight main steps, each of which is a potential target for restriction factors. First, HIV-1 attaches and fuses with the host cell membrane, and the viral core, i.e., the viral capsid containing two single strands of RNA, the enzymes reverse transcriptase, integrase and protease, and additional variable viral proteins, is released into the cytoplasm. Subsequently, upon uncoating of the viral core, viral single stranded RNA is reverse transcribed into double stranded viral DNA, forming the pre-integration complex (PIC), which is imported into the nucleus. Upon integration of the viral DNA into the host DNA, HIV-1 utilizes the host transcription machinery to produce viral mRNA. Following nuclear export, these mRNAs serve as templates for viral proteins that are assembled into new viral particles at the cell surface. Finally, these viral particles bud and, upon cleavage by a viral protease, are released from the cell as mature infectious virus [28,29].

Several host restriction factors target the early stages of the viral replication cycle, prior to nuclear transport. For example, the constitutively expressed host restriction factor Serine Incorporator 5 protein (SERINC5), which inhibits the translocation of the viral capsid to the cytoplasm, restricts HIV-1 infection at an early stage of the replication cycle [30]. Similarly, several members of the interferon-induced transmembrane (IFITM) protein family, in particular IFITM2 and IFITM3, have been demonstrated to interfere with viral fusion [31,32]. In virus-producing cells, IFITMs are incorporated into daughter virions, which confers reduced infectivity [31,33]. Multiple restriction factors then target viral nucleic acids to prevent reverse transcription. For example, Apolipoprotein B mRNA Editing Enzyme Catalytic Subunit (ABOBEC) 3G accomplishes restriction by introducing guanosine-to-adenosine hypermutation of the viral plus-strand DNA, leaving the virus incapable of effective reverse transcription [34]. Meanwhile, the dGTP-dependent deoxynucleotide triphosphohydrolase SAM domain and HD domain-containing protein 1 (SAMDH1) functions by starving the viral reverse transcriptase of intracellular dNTPs via hydrolyzation of these intracellular nucleic acids in non-cycling myeloid and CD4^+^ T cells [35,36,37]. Myxovirus resistance protein B (MxB; also known as MX2) is a late post-entry restriction factor, which interferes with nuclear import after reverse transcription to inhibit integration of proviral DNA, although it has also been suggested that MxB may additionally bind the HIV-1 capsid and inhibit uncoating earlier in the viral replication cycle [38,39].

Later in the virus replication cycle, Schlafen11 (SLFN11) then acts by selectively inhibiting expression of viral proteins. SLFN11 binds transfer RNA (tRNA) non-specifically, leading to preferential inhibition of viral protein synthesis due to the higher level of rare codons in viral genes as compared to host genes [40]. Subsequently, at the final stages of the virus replication cycle, additional HIV-1 restriction factors interfere with viral protein processing or function. The recently discovered interferon-inducible guanylate binding protein 5 (GBP5) targets a late stage of the viral replication cycle by interfering with processing of HIV-1 envelope glycoproteins, ultimately leading to assembly of non-infectious HIV-1 virions [41,42]. Membrane-associated RING (really interesting new gene)-CH (MARCH)-2 and MARCH8, which are both members of a larger E3 Ubiquitin ligase family, prevent the formation of infectious virions by targeting viral envelope proteins for degradation or intracellular retention [25,43]. Finally, tetherin, also known as BST-2 or CD317, accomplishes late-stage restriction of HIV-1 by preventing release of Vpu-deleted HIV-1 from restrictive cells, resulting in accumulation of mature HIV-1 particles on the cell surface [23,44].

### 1.2. TRIM5α: From HIV-1 Restriction Factor to Induction of Antiviral Immunity

Recent literature highlights that some restriction factors can be doubly functional upon sensing HIV-1, to both interfere with viral replication and also to act as sensors that initiate innate immune signaling pathways, which help to prevent further viral dissemination throughout the host [45,46,47,48,49].

The alpha isoform of tripartite motif (TRIM) protein 5 (TRIM5α) is one such restriction factor with both virus-restricting and -sensing properties. TRIM5α was first described as a restriction factor upon discovery that it potently inhibited HIV-1 infection in rhesus macaques [50]. Already in the late 1990s, data emerged to demonstrate a species-specific block of HIV-1 prior to reverse transcription occurring in Old World Monkeys, but not humans [51,52,53,54,55]. However, the responsible species-specific dominant repressive factor targeting the incoming HIV-1 capsid remained undiscovered until 2004. Stremlau and colleagues identified TRIM5α as the post-entry host factor potently restricting HIV-1 [50]. Expression of rhesus macaque TRIM5α (rhTRIM5α) in HeLa cells led to significantly decreased HIV-1 infection as compared to HeLa cells containing empty vectors. However, although expression of rhTRIM5α in HeLa cells efficiently blocked HIV-1 infection, simian immunodeficiency virus (SIV) was less restricted [50]. Furthermore, ectopic expression of the human TRIM5α orthologue (huTRIM5α) failed to restrict HIV-1 as potently as rhTRIM5α. Taken together, these data led to the conceptualization of TRIM5α as a species-specific HIV-1 restriction factor, with rhTRIM5α but not huTRIM5α efficiently restricting the virus [50]. In the following years, a wealth of literature has since focused on describing this TRIM5α-mediated mechanism of restriction [49,56,57,58,59,60,61,62,63].

It has since been demonstrated that rhTRIM5α restricts HIV-1 infection by forming hexagonal nets around incoming virus capsids and subsequently directing these TRIM5α-HIV-1 complexes for degradation [56,57,58,59,60]. More recently, rhTRIM5α was also reported to act as a pathogen recognition receptor (PRR) by sensing incoming retroviral capsids, and upregulating IFN responses as well as AP-1 and NF-κB-responsive inflammatory mediators [49,61,62,63]. Notably, recent literature indicates that huTRIM5α is also a HIV-1 restriction factor; however, it functions via a distinct mechanism as compared to rhTRIM5α, and it likely operates in a cell-specific manner [62,64,65]. huTRIM5α antiretroviral activity is also regulated by supplementary host proteins, such as cell-specific receptors and chaperones [64,66].

Here, we will explore the role of TRIM5α in both HIV-1 restriction and antiviral immunity, with a focus on human TRIM5α. We will emphasize the relevancy of huTRIM5α as a restriction factor, review its described restriction mechanisms and associated cellular degradation pathways, and discuss the role of TRIM5α in informing the host innate immune system. Finally, we will outline potential applications of harnessing TRIM5α-mediated antiviral pathways in innovative HIV-1 therapies.

## 2. TRIM5α-Mediated HIV-1 Restriction Across Species

### 2.1. TRIM5α of Old World Monkeys: The Case of Rhesus Macaques

rhTRIM5α is an approximately 500-amino-acid-long protein that exists as either a free cytoplasmic protein or localized within highly mobile cytoplasmic clusters termed cytoplasmic bodies [50]. It is a member of the extensive TRIM (tripartite motif) protein family, so named for their inclusion of three characteristic domains: an amino-terminal RING (really interesting new gene) domain, a B-box, and a coiled-coil (CC) region [67]. The RING and B-box domains are connected via the linker region 1 (L1). This TRIM motif is connected to a carboxyl-terminal SPRY (also known as B30.2) domain via the flexible linker region 2 (L2). The SPRY domain, and in particular its variable sequence 1 (v1), one of three variable peptide sequences in the SPRY domain, mediates TRIM5α interaction with retroviral capsids (Figure 1A) [68,69].

Studies conducted using purified rhTRIM5α protein preparations or in human (HeLa) or canine (Cf2Th) cell lines stably transfected with rhTRIM5α have found that the RING, SPRY, CC, and B-box domains of rhTRIM5α all contribute to the HIV-1 restriction process. The CC and B-box domains facilitate the initial dimerization and subsequent multimerization of rhTRIM5α, promoting its aggregation into cytoplasmic bodies and, upon HIV-1 infection, formation of hexagonal rhTRIM5α nets around target virus capsid proteins [58,59,60]. *In vitro* mutations of the B-box residue R121 inhibit the formation of hexagonal structures on the surface of HIV-1 capsid, and thereby abrogated rhTRIM5α-mediated HIV-1 restriction (Figure 1A) [58,59,70]. In addition, an I193A mutation within the CC-domain of a rhTRIM5α fusion protein resulted in loss of restriction in HeLa cells and a slight instability of the rhTRIM5α dimer [71]. Structural modeling indicated that residue I193 is likely important for the correct packaging of the CC/L2/SPRY domains, and that the I193A mutation may alter the positioning of the SPRY domain relative to the CC-domain, which is associated with defective binding to the viral capsid [71].

Cryogenic electron microscopy studies with purified rhTRIM5α protein preparations have demonstrated that binding of the SPRY domain to the virus capsid promotes formation of hexagonal rhTRIM5α nets that physically mirror the viral capsid surface lattice, thus compensating for the low affinity of the SPRY domain for the viral capsid by increasing binding avidity, and reinforcing rhTRIM5α binding efficiency [60,72,73]. In HeLa and 293T cell lines transfected with rhTRIM5α, interaction of rhTRIM5α with viral capsid cylinders leads to structural disruption of the HIV-1 capsid proteins, driving premature uncoating of the virus and thereby inhibiting viral genome translocation to the nucleus and retroviral integration [50,74,75].

The E3 ubiquitin ligase activity of the RING domain of rhTRIM5α is key in directing rhTRIM5α-mediated degradation of the HIV-1 capsid. Demonstrative of this, the R60A mutation within the rhTRIM5α RING domain, which abolishes its E3 ubiquitin ligase activity, interferes with retroviral restriction (Figure 1A) [76,77]. The rhTRIM5α RING domain normally directs the elongation of N-terminally anchored K63-linked ubiquitin chains to the viral capsid, which tags the incoming virus for destruction. These ubiquitin-tagged rhTRIM5α-HIV-1 complexes are then directed to the proteasome for subsequent degradation [56,78,79,80].

It has been reported that proteasome inhibition with small-molecule inhibitor MG132 or deletion of the RING domain limits but does not completely abrogate rhTRIM5α-mediated HIV-1 restriction [78,81]. Proteasome inhibition or introduction of RING domain mutations C15A or C18A in rhTRIM5α alters the intracellular localization of rhTRIM5α, causing it to accumulate in relatively large cytoplasmic or (peri)nuclear bodies, respectively, and resulting in decreased availability of rhTRIM5α to restrict HIV-1 [78,81]. Furthermore, disrupting proteasome function permitted the generation of HIV-1 late reverse transcription products, although infection with a single cycle R7ΔEnvGFP reporter HIV-1 virus, as measured by GFP^+^ cells, remained impaired. Taken together, these data underline that rhTRIM5α acts soon after entry of the retroviral capsid into the cytoplasm, prior to reverse transcription, and together with the proteasome system prevents reverse transcription of HIV-1 and drives proteasomal degradation of rhTRIM5α-HIV-1 complexes.

### 2.2. TRIM5α of New World Monkeys: The Case of TRIMCyp

It is common amongst Old World monkeys such as Rhesus macaques to exhibit TRIM5α-mediated post-entry restriction of HIV-1, but this phenotype is less common amongst New World monkeys. Owl monkeys specifically exhibit a distinct pattern of HIV-1 restriction not detected in the other New World monkeys such as squirrel monkeys, golden-headed lion tamarins, or black-tailed marmosets [51]. The protein responsible for HIV-1 restriction in owl monkeys was identified to be a fusion protein in which the SPRY domain of TRIM5α has been replaced by the coding sequence of Cyclophilin A (CypA) due to a historical Long Interspersed Element 1 (LINE-1) retrotransposition event [82]. CypA is a host chaperone prolyl isomerase that has interacted with prehistoric lentivirus capsids related to HIV-1 throughout evolution via an exposed peptide loop on the viral capsid lattice [83,84]. The resultant fusion protein, termed TRIMCyp, is a potent inhibitor of lentiviruses, including of HIV-1 by owl monkeys [60,82,85,86]. In addition, CypA has been implicated in MxB-mediated HIV-1 restriction, although the molecular details of this interaction are not yet fully elucidated [87,88].

Owl monkey TRIMCyp-mediated HIV-1 restriction relies on multimeric binding to incoming virus, with the CypA domain (replacing the SPRY domain of rhTRIM5α) binding the HIV-1 capsid [89,90]. Recent studies have highlighted that, similar to rhTRIM5α, the TRIMCyp CC and B-box domains facilitate its dimerization and subsequent multimerization, thereby promoting aggregation into cytoplasmic bodies and, upon HIV-1 infection, formation of hexagonal TRIMCyp nets around target virus capsid proteins [60,91]. Higher-order assembly of TRIMCyp may serve an analogous function of coupling capsid binding and ubiquitination to promote HIV-1 degradation, as with rhTRIM5α.

Like rhTRIM5α, mutating amino acid residue R120 within the TRIMCyp B-box domain abrogates HIV-1 restriction by TRIMCyp (Figure 1A) [70]. In contrast to the rhesus orthologue, mutation of the TRIMCyp CC-domain residue I192 did not result in the abrogation of HIV-1 restriction [71].

With regard to capsid binding, N369 within the CypA domain of TRIMCyp is a key residue for HIV-1 capsid recognition (Figure 1A) [92]. Correspondingly, treatment of HIV-1-GFP-infected HeLa cells with cyclosporine A, which binds the CypA domain, completely abrogated TRIMCyp restriction, indicating that viral capsid binding by the CypA domain is essential for the TRIMCyp’s HIV-1 inhibitory function [91,93].

### 2.3. Human TRIM5α

Akin to its rhesus macaque orthologue, the huTRIM5α structure contains the characteristic RING, B-box, CC region, and SPRY domains. Exogenous huTRIM5α is able to restrict some lentiviruses, such as N-tropic murine leukemia viruses (N-MLV) in human and mouse cell lines [94]. Unlike rhTRIM5α, huTRIM5α does not efficiently restrict HIV-1 in cell lines [50]. For example, huTRIM5α stably expressed in HeLa cells only slightly inhibits HIV-1-GFP (recombinant HIV-1 expressing GFP pseudotyped with VSV-G glycoprotein) when compared to HeLa-cells expressing rhTRIM5α [50]. Likewise, huTRIM5α expressed in the feline cell line CRFK weakly restricts HIV-1, while rhTRIM5a completely blocks infection [95]. It has previously been suggested that the differing patterns of retrovirus restriction by rhTRIM5α versus huTRIM5α observed in cell lines is due to variation in the SPRY v1 sequence between the two orthologues, in line with the long-standing belief that HIV-1 restriction by rhTRIM5α is species-specific [96,97].

The apparent decreased ability of huTRIM5α to restrict HIV-1 in cell lines has been associated to low affinity for the viral core and instability of the protein [98]. In support of this, a single amino acid substitution on position 332 from arginine to proline (R322P), mimicking the sequence in rhTRIM5α, conferred the ability of huTRIM5α to restrict HIV-1, though ten times less efficient than rhTRIM5α [99]. Furthermore, a R332/R335 double mutation in the SPRY domain provided HIV-1 resistance similar to rhTRIM5α in a T-cell line (SUP-T1) and in primary human T-cells [97,98]. In addition, stabilization of huTRIM5α via fusion to mCherry resulted in robust restriction of HIV-1 strains SF162 and BK132 [98]. This indicated that huTRIM5α is able to bind HIV-1 but that the increased degradation of the human orthologue as compared to rhTRIM5α hampers the HIV-1 restrictive capabilities of huTRIM5α. Furthermore, it has also been demonstrated that the huTRIM5α B-box residue R119, similar to its orthologues rhTRIM5α (R121) and TRIMCyp (R120), is essential for hexagonal structure formation and viral restriction [70]. huTRIM5α R119E and R119D mutants lost antiretroviral activity against non-human viruses such as N-MLV (Figure 1A) [70].

Notably, Ribeiro and colleagues were the first to show that huTRIM5α is a potent HIV-1 restriction factor in specific primary human cells [64]. However, it functions via a distinct mechanism to rhTRIM5α, which relies on HIV-1 uptake via specific cell surface receptors and on an alternative cellular degradation system in human primary immune cells [64]. Recent literature support these findings and further demonstrated the capacity of huTRIM5α to bind the HIV-1 capsid and efficiently restrict HIV-1 infection in human primary immune cells using alternative host proteins and cellular degradations pathways [65,66,100].

## 3. The Nexus Between TRIM5α Antiviral Functions and Host Degradation Pathways

Abnormal, aggregated, or non-self-cytosolic material is typically degraded by one of two host degradative pathways: the proteasome system or the autophagy pathway [101,102]. Both systems are capable of degrading ubiquitinated intracellular substrates. Differential ubiquitination and the oligomeric state of the receptors that recognize the substrate destined for degradation determine which degradative pathway the substrate will take. Monomeric proteasome receptors have high affinity for ubiquitin and therefore preferably bind soluble substrates. On the other hand, substrate aggregation favors autophagic degradation due to the high avidity of oligomeric receptors [101,103].

The proteasomal degradation system relies on three categories of ubiquitin-related enzymes, namely, E1 ubiquitin-activating enzymes, E2 ubiquitin-conjugating enzymes, and E3 ubiquitin ligases [104]. First, an E1 ubiquitin-activating enzyme binds ubiquitin molecules and transfers the molecule to an E2 ubiquitin-conjugating enzyme. Then, the E2 ubiquitin-conjugating enzyme, together with the bound ubiquitin, interacts with the substrate-bound E3 ubiquitin ligase [105]. RING-type E3 ubiquitin ligases, so named for their inclusion of a RING domain, as does TRIM5α, determine the fates of cytosolic target proteins by recognizing and transferring ubiquitin to target proteins. Repeated interactions of these three enzymes results in polyubiquitination of the substrate, generally forming a so-called K48 ubiquitin chain, i.e., a polymeric ubiquitin chain formed via linking of the K48 residues of multiple ubiquitin monomers, thereby earmarking the substrate for degradation via the 26S proteasome [106,107]. The 26S proteasome consists of a 20S proteolytic core particle that is flanked by two 19S lid particles. Upon recognition of the ubiquitinated substrate by the 19S lid, the substrate is translocated into the barrel shape of the proteasome and cleaved by proteases in the catalytic sites of the 20S core [105,108]. Proteasomal degradation of HIV-1 proteins may have both proviral and antiviral roles as reviewed in [109], including rhTRIM5a-mediated HIV-1 degradation.

Autophagy is a process in which specialized double-membrane intracellular vesicles form to engulf cytosolic cargo such as viral components and target them for lysosomal degradation [110]. Thus, autophagy functions as an innate immune defense mechanism against intracellular pathogens. Generally, larger cytoplasmic cargo that exceeds the spatial capacity of proteasomes is directed to autophagy-mediated degradation [111]. The initiation of the autophagosomal membrane, which sequesters cytosolic cargo, is under control of several proteins, including ULK1 and Beclin 1 [112,113]. After formation of the autophagosome, which requires the involvement and cooperation of multiple proteins including autophagy-related (ATG) proteins, further elongation of the nascent autophagic membrane is dependent on ubiquitin-like conjugation complexes. The ubiquitin-like protein ATG12 forms a complex with ATG5 and ATG16L1, that together with the proteins ATG4, ATG7, and ATG3, processes and lipidates the ubiquitin-like microtubule-associated protein 1 light chain 3 (LC3) into LC3-II [114]. LC3-II is subsequently conjugated to membrane-embedded phosphatidylethanolamine (PE) in the inner and outer membrane of the autophagosome, via the E3 ligase activity of the ATG16L1/ATG12/ATG5 complex, and directs sequestering of cargo including an LC3-interacting region (LIR) [115,116,117]. Additionally, LC3-ll is essential for elongation and closure of the phagophore membrane to form the autophagosome [118]. Subsequently, mature autophagosomes fuse with lysosomes resulting in the formation of autolysosomes, within which the cytoplasmic cargo is degraded due to acidification of the autolysosomal lumen and the activity of lysosomal hydrolases [119]. This continual process of degradation can be referred to as autophagy flux [120].

Selective autophagy, which can be differentiated from bulk autophagy by its specific targeting of cytoplasmic components including protein aggregates and intracellular pathogens for degradation, is dependent on interactions between autophagy receptors, LC3, and cargo [121,122]. Autophagy receptors such as p62 contain a ubiquitin binding site as well as a LIR motif. Thus, these receptors simultaneously bind to poly-ubiquitinated cytoplasmic cargo, and to LC3-II that coats the autophagosomal membrane, thereby directly targeting cargo into autophagosomes [121,123]. Notably, TRIM5α has been demonstrated to include a LIR motif in its CC domain, and to bind p62 [26,124]. Additionally, TRIM proteins can induce a similar but distinct process termed precision autophagy. Here, autophagy can be induced by TRIM proteins bound to cargo without requiring ubiquitination or other protein tags [125].

### 3.1. Cellular Degradation Pathways in HIV-1 Restriction: The Proteasome Versus Autophagy

The main mechanism underlying the restrictive capacity of TRIM5α was long suggested to be early disruption of the HIV-1 capsid and, subsequently, capsid degradation via the proteasome system (Figure 1B) [56,78]. rhTRIM5α-mediated, proteasome-dependent HIV-1 restriction was originally discovered upon transduction of rhTRIM5α in HeLa cells incubated with single round recombinant GFP-reporter HIV-1 pseudotyped with VSV-G [50]. Expression of rhTRIM5α in these cells led to significantly decreased infection rates as compared to inoculation of parental HeLa cells. The E3 ubiquitin ligase activity of the rhTRIM5α RING domain facilitates proteasome-dependent HIV-1 restriction. Together with the RING domain, the B-box 2 and CC domains promote formation of higher order complexes of rhTRIM5α, which enhances the interaction of the SPRY domain with the viral capsid of HIV-1, as described in Section 2.1 in more detail [126]. Subsequently, the RING domain initiates polyubiquitination of rhTRIM5α-HIV-1 complexes located within cytoplasmic bodies [127].

Co-localization of rhTRIM5α with the proteasomal system has also been demonstrated in human 293T and HeLa cell lines stably expressing rhTRIM5α [57]. Notably, the immunoproteasome has recently been implicated in IFN-facilitated, huTRIM5α-mediated HIV-1 restriction in CD4^+^ T cells, as is discussed in more detail in Section 4.3 [65,100]. However, there have been conflicting reports with regard to the significance of proteasomal activity in rhTRIM5α-mediated HIV-1 restriction. rhTRIM5α and owl monkey TRIMCyp are still active in presence of proteasome inhibitors or in cell lines lacking active ubiquitin-activating E1 enzymes [56,128]. Additional studies have indicated that although proteasome inhibition with MG132 prevents rhTRIM5α-mediated capsid disassembly and restores HIV-1 reverse transcription in HeLa cells, HIV-1 infection is still impaired [56,78]. This suggests either a two-phase restriction mechanism of rhTRIM5α, involving first a disruption in passage of the viral genome to the nucleus and secondly targeting of rhTRIM5α-HIV-1 complexes for degradation, or the involvement of additional degradation pathways [24,78]. In support of the latter possibility, it has recently been demonstrated that both rhTRIM5α and huTRIM5α can act as platforms for assembly of autophagy regulating proteins, indicating that, similar to other TRIM proteins, TRIM5α can induce precision autophagy pathways [64,124,129].

### 3.2. TRIM-Mediated Precision Autophagy

As briefly described above, TRIM proteins have been shown to mediate an additional form of autophagy, termed precision autophagy [125,129,130]. Precision autophagy is distinct from selective autophagy, which typically requires at least two different host proteins—one for target recognition and a second for promoting assembly of the autophagy machinery [64,125,129,131,132]. In contrast, during TRIM-mediated precision autophagy, TRIM proteins play dual roles: they both directly recognize target cargo and assemble the proteins required for autophagy, including initiator proteins (e.g., ULK1 and Beclin 1), elongation proteins (e.g., ATG5), and LC3.

In HeLa cells, both huTRIM5α and rhTRIM5α were demonstrated to interact with LC3 via the helical LIR within their CC domains [124,129]. Furthermore, huTRIM5α and rhTRIM5α were demonstrated to interact with ULK1 via their SPRY domains, thereby inducing autophagy [129]. In addition, knockdown of autophagy proteins Beclin 1 and p62 in rhesus FRhK-4 and rhesus primary CD4^+^ T cells has been demonstrated to affect, although not abrogate, rhTRIM5α-mediated restriction of HIV-1 and VSVG-pseudotyped HIV-1 [129,133]. Notably, knockdown of ATG16L1 in a rhTRIM5α-transduced U87 cell line also slightly reduced HIV-1 restriction by rhTRIM5α [64]. Furthermore, exogenous rhTRIM5α was recently demonstrated to be localized within different autophagy vesicles in HeLa cells, although in a second study, depletion of ATG5 and Beclin 1 in HeLa cells transduced with rhTRIM5α did not result in significantly decreased restriction of VSV-g pseudotyped HIV-1 reporter virus infection [133,134]. Taken together, these results indicate that rhTRIM5α can act as a platform for the assembly of active autophagy components and formation of autophagy vesicles upon sensing the HIV-1 capsid in both human and macaque cell lines, although the autophagy pathway, in contrast to proteasome system, appears to have minimal importance in the rhTRIM5α-mediated HIV-1 restriction mechanism.

Strikingly, in specific primary human cell subsets, huTRIM5α has been demonstrated to not only act as an efficient scaffolding platform for autophagy regulators upon HIV-1 sensing, but also to restrict HIV-1 via an autophagy-dependent mechanism. These findings underline a pivotal role for autophagy in HIV-1 infection of human primary cells [64]. Upon recognizing its cargo, huTRIM5α has been shown to form complexes with ATG16L1 and ATG5, thereby promoting degradation of its bound cargo via precision autophagy as illustrated in Figure 1B [64]. Silencing of ATG5 or ATG16L1 in primary human Langerhans cells (LCs) resulted in increased HIV-1 integration, underlining the role of autophagy in huTRIM5α-mediated HIV-1 restriction [64]. Although the ability of TRIM5α to form scaffolding platforms for assembly of autophagy regulating proteins is conserved throughout primate evolution, autophagy primarily plays a pivotal role in huTRIM5α autophagy-mediated HIV-1 restriction, unlike the apparent lesser role of autophagy in HIV-1 restriction by its rhesus orthologue [64,125,135].

### 3.3. Human TRIM5α Restricts HIV-1 in Epithelial Langerhans Cells

In the late 2000s, it was discovered that primary human LCs, a subset of Dendritic cells (DCs) that reside in mucosal epithelium, are a natural barrier to HIV-1 infection (Figure 2A,B) [64,136,137]. Together with subepithelial DCs, epithelial LCs are among the first immune cells to come in contact with HIV-1 during sexual transmission [138,139]. The HIV-1 receptor for LCs is the cell-specific C-type lectin Langerin, which allows LCs to efficiently capture HIV-1 from the extracellular milieu [136,140,141]. Upon capture of HIV-1, Langerin directs internalization of HIV-1 virions to Birbeck granules, which are LC-specific organelles that play a key role in LC-specific restriction of HIV-1 [136,137]. However, the mechanism of HIV-1 degradation after internalization into Birbeck granules in LCs remained a mystery for nearly a decade. The restriction was not facilitated by the proteasomal system, as demonstrated by MG132 not interfering with HIV-1 restriction [64]. Notably, HIV-1 restriction by Langerin occurs after HIV-1 fusion but before integration of viral DNA into the host genome [142]. It has been reported that the post-entry restriction factor SAMHD1 is ineffective in LCs, indicating that another post-entry antiviral factor must be of importance [143,144].

Notably, in primary human LCs, CD4^+^CCR5^+^Langerin^+^U87 cells, and MUTZ3-derived LCs infected with CXCR4- or CCR5- tropic viruses, RNAi-based silencing of huTRIM5α resulted in increased HIV-1 integration into the host genome, increased productive infection, and enhanced HIV-1 transmission to CD4^+^ T cells [64]. huTRIM5α mediates the assembly of an autophagy-activating scaffold to Langerin, which targets the HIV-1 capsid for autophagic degradation and prevents infection of LCs. We were therefore the first to show that huTRIM5α is a potent HIV-1 restriction factor in LCs. At steady-state, Langerin associates with the LSP-1–TRIM5α–ATG16L1 complex. Capture of HIV-1 by Langerin then targets internalization of the incoming virus into Birbeck granules. Upon viral fusion, huTRIM5α mediates recruitment of ATG5 to the TRIM5α–ATG16L1–HIV-1p24 capsid complex, which promotes induction of autophagosome formation and subsequently targeting of HIV-1 capsid complexes into autophagy vesicles for lysosomal degradation (Figure 2B) [64,145,146].

Furthermore, silencing of either huTRIM5α or the autophagy regulating proteins ATG5 or ATG16L1 in primary human LCs and CD4^+^CCR5^+^Langerin^+^U87 cells abrogated HIV-1 restriction, while treatment with the autophagy-enhancing drug rapamycin decreased HIV-1 integration in primary human LCs [64]. CD4^+^CCR5^+^U87 cells overexpressing huTRIM5α demonstrated increases in both ATG5 recruitment to ATG16L1-TRIM5α complexes, and in LC3-II levels in the presence of bafilomycin, indicating that huTRIM5α overexpression boosts autophagy levels [64]. Altogether, these data show that HIV-1 degradation in primary human LCs occurs via a TRIM5α-directed autophagy restriction mechanism.

Notably, huTRIM5α-mediated HIV-1 restriction is strongly dependent on the viral uptake pathway. Although huTRIM5α efficiently restricts HIV-1 in primary human LCs, it does not restrict VSV-G-pseudotyped HIV-1, which bypasses Langerin in both primary LCs as well as CD4^+^CCR5^+^Langerin^+^U87 cells [64,147]. In contrast, rhTRIM5α efficiently restricts both HIV-1 and VSV-G-pseudotyped HIV-1 [50,64,75].

Primary human DCs, which do not express Langerin but rather the C-type lectin receptor DC-SIGN, are more permissive to HIV-1 (Figure 2C) [64,136,148,149,150]. Unlike Langerin, DC-SIGN facilitates HIV-1 infection of and transmission by DCs, and silencing huTRIM5α in DC-SIGN^+^ DCs did not affect levels of HIV-1 integration nor infection [64,151,152]. Whereas huTRIM5α remains in association with the Langerin signalosome upon HIV-1 capture by Langerin in LCs, binding of HIV-1 to DC-SIGN in DCs leads to disassociation of huTRIM5α from the DC-SIGN signalosome, possibly contributing to the lack of huTRIM5α-mediated HIV-1 restriction in submucosal DCs (Figure 2B,C) [64]. Furthermore, HIV-1 rapidly downregulates autophagy in DCs, which may represent an additional strategy evolved by the virus to evade autophagy-mediated degradation [64,153]. Thus, these data outline a novel autophagy- and receptor- controlled huTRIM5α restriction mechanism, dictating protection or infection of human DC subsets.

The restrictive capacity of LCs is impaired with specific HIV-1 variants, and during conditions such as tissue immune activation and co-infections, resulting in increased susceptibility to HIV-1. For example, some Transmitted/Founder HIV-1 strains are able to escape Langerin-dependent huTRIM5α-mediated HIV-1 restriction in LCs [154]. Furthermore, co-infections, for example, with additional sexually transmitted infections such as gonorrhea and candidiasis, also increase LC susceptibility to HIV-1 infection and transmission [155,156,157]. Lastly, activation of LCs by tumour necrosis factor (TNF) can also promote HIV-1 transmission from LCs to T-cells [155]

## 4. HIV-1 Escapes huTRIM5α-Mediated Restriction in Submucosal DC-SIGN^+^ Dendritic Cells and CD4^+^ T Cells

### 4.1. Submucosal Dendritic Cells

Although huTRIM5α has been demonstrated to restrict HIV-1 in an autophagy-dependent manner in primary human LCs, several groups have independently shown that huTRIM5α is unable to restrict HIV-1 in other primary targets such as submucosal DCs [64,153]. An important receptor for HIV-1 transmission is the highly expressed surface receptor DC-SIGN [152]. Normally, upon binding to a ligand, DC-SIGN facilitates receptor-mediated endocytosis to initiate antigen-processing [158]. In addition, DC-SIGN facilitates interaction of DCs with CD4^+^ T-cells via the immunological synapse [152]. Several studies have also shown that submucosal DCs are able to facilitate HIV-1 transmission via a mechanism termed *trans*-infection [152,159,160]. During this mechanism, DCs capture the virus and transmit it to CD4^+^ T-cells via a virological synapse, without replication of the virus within the DCs. Several independent studies have also reported productive HIV-1 infection of submucosal DCs, with active replication prior to transmission to other cells (termed *cis*-infection), although productive infection of DCs compared to CD4^+^ T-cells is 10- to 100-fold lower [150,151,161]. In the case of productive infection, DC-SIGN strongly binds the HIV-1 envelope glycoprotein gp120 and together with other PRRs is essential for viral replication in DCs [151,162,163]. In particular, DC-SIGN-mediated signaling, which is induced upon binding of the HIV-1 envelope glycoprotein gp120, is necessary for eventual generation of full-length virus transcripts [151]. In addition, HIV-1 envelope protein Env activates mammalian target of rapamycin (mTOR), thereby downregulating autophagy, which prevents autophagy-mediated viral degradation. In support of this, treatment with the mTOR inhibitors everolimus or rapamycin results in induction of autophagy, leading to a reduction in productive HIV-1 infection of tissue-derived DCs, and a decrease in transmission of HIV-1 by human DCs to CD4^+^ T-cells [153,164].

Notably, at steady-state conditions, huTRIM5α forms a complex with DC-SIGN, in a manner similar to that with Langerin as described in Section 3.3 [64]. However, upon binding of HIV-1 to DC-SIGN, huTRIM5α dissociated from the DC-SIGN signalosome [64] (Figure 2C). In concurrence with this data, silencing of huTRIM5α in DC-SIGN^+^ DCs had no effect on HIV-1 integration or infection [64]. This highlights that multiple C-type lectin receptors, including DC-SIGN and Langerin, can form complexes with huTRIM5α, but that the HIV-1 restriction properties of huTRIM5α are cell-specific and receptor-dependent.

Recently, new light was shed on the lack of HIV-1 restriction by huTRIM5α in primary human monocyte-derived DCs [62,144]. In human DCs, huTRIM5α is sequestered in Cajal bodies, a subset of membrane-less nuclear bodies consisting mostly of proteins and RNA, and other nuclear bodies via a small ubiquitin-like modifier (SUMO)ylation-dependent manner. Nuclear sequestration of huTRIM5α makes room for innate immune sensing of retroviral genetic material by the DNA sensor cGAS in the cytosol, leading to potent induction of type I IFN responses that are key for promoting an antiviral state [62,144]. Strikingly, overexpression of huTRIM5α in human DCs, or treatment with an inhibitor of SUMOylation, restored cytoplasmic huTRIM5α expression and efficient huTRIM5α-mediated HIV-1 restriction [62]. These results suggest the potentiality of huTRIM5α to restrict HIV-1 in other DC subsets in addition to LCs.

Furthermore, a recent study demonstrated the potential role for CypA in abrogating huTRIM5α-mediated HIV-1 restriction in DCs [165]. Infection of primary human monocyte-derived DCs with single round VSV-G pseudotyped HIV-1 viruses bearing capsid mutation P90A, which prevents CypA from interacting with the capsid, was less efficient as compared to infection with viruses lacking the P90A mutation [66,166,167]. HIV-1 infection by HIV-1 P90A was restored upon depletion of huTRIM5α in the DCs, which further highlights the capacity of huTRIM5α to inhibit HIV-1 in DCs [165]. It is important to note here that VSV-G-pseudotyped HIV-1 largely bypasses the DC-SIGN uptake route, and that the observed huTRIM5α-mediated restriction of VSV-G pseudotyped HIV-1 P90A in DCs upon CypA depletion is yet to be confirmed with HIV-1 bearing the viral glycoprotein gp120 [165,168].

Notably, it was also recently demonstrated that rhTRIM5α does not restrict HIV-1 in primary rhesus macaque monocyte-derived DCs, unlike in other rhesus macaque primary cells or cell lines, or in human cell lines ectopically expressing rhTRIM5α [50,169]. Rather, rhTRIM5α was SUMOylated and sequestered in the nucleus of rhesus macaque DCs following the same mechanism described in human DCs. Strikingly, this lack of restriction by rhTRIM5α was also demonstrated in cynomolgus and pigtailed macaque DCs, as well as in African green monkey DCs, indicating that lack of TRIM5α-mediated restriction is prevalent in humans and other primate DCs [62].

This lack of TRIM5α-mediated restriction in DCs is thus conserved throughout evolution, possibly as a tradeoff for viral sensing and induction of type I IFN responses. Restoration of huTRIM5α-mediated restriction in DCs by blocking host pathways such as CypA binding and SUMOylation provides insight into novel therapeutic possibilities.

### 4.2. Macrophages

Macrophages are myeloid lineage cells that are resident in subepithelial tissues throughout the body. They can originate from self-renewing tissue-resident macrophages or infiltrating blood-derived monocytes [170,171]. As antigen-presenting cells, macrophages are important for bridging innate and adaptive immunity by phagocytosing extracellular pathogens, and presenting pathogen-derived antigens to CD4^+^ T cells [172]. As macrophages express the HIV-1 entry receptor CD4 and co-receptors CCR5 and CXCR4, they are themselves susceptible to HIV-1 infection [13]. In addition, HIV-1 infected macrophages can transmit virus to CD4^+^ T cells via a viral synapse, i.e., cell-to-cell contact, in a manner similar to that of DCs (Figure 2D) [173,174,175].

In macrophages, HIV-1 hijacks the early non-degradative stages of autophagy. Research using primary human macrophages, as well as THP-1 and U937 monocyte-derived macrophages, has demonstrated that the induction of autophagy with rapamycin increased extracellular HIV-1 yields [176]. In addition, inhibition of autophagy via treatment with 3MA or knock-out of Beclin1 and ATG7 resulted in a reduction in the extracellular release of infectious virus as compared to untreated cells, indicating that autophagy promotes HIV-1 maturation or release in macrophages [176]. This is in contrast with other immune cell types, as induction of autophagy in LCs, DCs, and CD4^+^ T cells results in a decrease in production of mature virions [64,164,177].

In macrophages, the interaction of toll-like receptor 8 (TLR8) and HIV-1 induces autophagy by Beclin 1-dependent dephosphorylation of transcription factor EB (TFEB) and subsequently its nuclear translocation during the initial stages of HIV-1 infection [178,179]. Upon the induction of autophagy, HIV-1 Gag proteins co-localize with autophagy-related protein LC3 and subsequently promote Gag processing [176].

The HIV-1 Nef protein plays an important role in manipulation of host autophagy, and thereby immune evasion, in macrophages. Several studies have indicated that, by inhibiting Beclin 1 association with class III PI 3-kinase (PI3KC3) complex II, which plays a key role in autophagosome-lysosome fusion, HIV-1 Nef blocks autophagosome maturation [176,180]. Interestingly, recent studies have also indicated that Nef additionally blocks autophagy at its early initiation stages [176,178,180,181].

In addition to evading host antiviral mechanisms in macrophages, HIV-1 also exploits host proteins to evade restriction. Recent studies have demonstrated that multimeric binding of host CypA proteins to the viral capsid prevents interaction of huTRIM5α with the viral core, thereby blocking huTRIM5α-mediated restriction [165,182]. Correspondingly, HIV-1 viruses bearing capsid mutation P90A, which prevents CypA from interacting with the capsid, were restricted by huTRIM5α in macrophages, as was also demonstrated in DCs [66,165,166,167]. Likewise, a construct generated by fusion of CypA to huTRIM5α (hT5Cyp), inspired by the New World monkey TRIMCyp as described in Section 2.2, was able to block HIV-1 infection in human primary macrophages [19].

### 4.3. T Cells

CD4^+^ T cells are key HIV-1 target cells that can be infected directly within the mucosal environment, or following transmission from DCs (Figure 2E) [144]. In addition, HIV-1 can latently persist in long-lived CD4^+^ T cells in PLWH [6,148,183]. As in DCs, HIV-1 infection inhibits autophagy in human CD4^+^ T cells, via HIV-1 Vif interactions with LC3, and a HIV-1 Nef-dependent block in autophagy flux resulting in accumulation of endosomes and lysosomes [184,185]. Moreover, as in macrophages, multimeric binding of host CypA to the incoming viral capsid blocks the interaction of huTRIM5α with the viral core, thereby preventing huTRIM5α-mediated restriction during CD4^+^ T cell infection. In addition, it was demonstrated recently that endogenous huTRIM5α in primary human CD4^+^ T cells is able to bind the HIV-1 capsid post-fusion, in the absence of CypA, and thereby restrict HIV-1 [66]. In addition, HIV-1 viruses bearing capsid mutations such as P90A and G89V, which prevent CypA interacting with the capsid, were unable to infect primary human CD4^+^ T cells [66,166,167]. Similar to in submucosal DCs and macrophages, infection by HIV-1-P90A was restored upon depletion of huTRIM5α in primary human CD4^+^ T cells or in Jurkat cells [66,165]. However, it is still unclear whether autophagy machinery plays a role in the huTRIM5α-mediated HIV-1 restriction in CD4^+^ T cells.

An additional IFNα-dependent mechanism for huTRIM5α-directed control of HIV-1 infection in human cell lines and primary human CD4^+^ T cells was recently demonstrated [65]. In primary human CD4^+^ T cells and in U87-MG.CD4^+^CXCR4^+^ cells transduced to ectopically express huTRIM5α, HIV-1 infection was efficiently inhibited in the presence of IFNα, in a manner dependent on SPRY domain-mediated recognition of the viral capsid [65]. The presence of IFNα stimulated activation of the immunoproteasome, a proteasome isoform primarily present in immune cells, and selectively promoted ubiquitination of huTRIM5α, thereby accelerating its proteolytic turnover, indicating that, in the presence of IFNα, the immunoproteasome may facilitate degradation of huTRIM5α-HIV-1 capsid complexes [65].

Prior to these findings, huTRIM5α had long been considered to be a poor HIV-1 restrictor in a variety of cell lines. We have shown for the first time that huTRIM5α is a potent restriction factor in human LCs, and that this restriction is dependent on autophagy mechanisms and cell-surface receptor usage by HIV-1. These recent studies further underline the capacity of huTRIM5α to potentially restrict HIV-1 in other relevant human target cells such as DCs, CD4^+^ T cells, and macrophages. Thus, efficient huTRIM5α restriction likely depends on TRIM5α subcellular localization (e.g., nucleus versus cytosol) within specific cell or tissue types, but also on the ability to induce autophagy- and immunoproteasome-facilitated degradation pathways, the presence of additional cellular factors that facilitate the action of huTRIM5α (e.g., Langerin), and the absence or inhibition of cellular factors that block huTRIM5α recognition of the viral capsid (e.g., CypA) [62,64,65,186].

### 4.4. Cell-Specific and Autophagy-Dependent Restriction of HIV-1 by Additional Restriction Factors

Cell-specific restriction is not exclusive to TRIM5α. For example, it has been suggested that differential SAMHD1 degradation may be an underlying contributor to the reduced permissiveness to HIV-1 in human macrophages and DCs as compared to activated CD4^+^ T cells. In support of this, pre-treatment with Vpx, a known inducer of SAMHD1 degradation during HIV-2 or SIV infection but absent in HIV-1, promoted degradation of SAMHD1 and enhanced susceptibility of macrophages and DCs to HIV-1 [35]. However, another study showed that SAMHD1 also protects resting CD4^+^ T cells in a similar manner to macrophages and DCs [187].

The restriction factor MARCH8 also shows cell-specific activity. MARCH8 mRNA levels have been demonstrated to be 5 to 8 times higher in primary human terminally differentiated myeloid cells versus CD4^+^ T cells and monocytes [25]. These expression levels aligned with decreased infection in differentiated myeloid cells versus high infection in CD4^+^ T cells. Taken together, these recent data suggest that in primary cells, HIV-1 restriction by different host antiviral factors is likely cell-specific, and modulated by the presence or absence, and expression or degradation levels of the restriction factors in question.

In recent years, several reports have also indicated that autophagy is involved in the activity of multiple HIV-1 restriction factors in addition to huTRIM5α. For example, tetherin has been shown to interact with autophagy via multiple different pathways. In the context of HIV-1 infection, HIV-1 Vpu has been reported to downregulate tetherin expression on the cell surface by promoting its ubiquitination and subsequent intracellular sequestration and/or degradation within autophagosomes or lysosomes, thus potentially hijacking the autophagy pathway for the benefit of the virus [188,189]. Strikingly, breast cancer-associated gene 2 (BCA2, also known as Rabring7, ZNF364, or RNF115), a RING-type E3 ubiquitin ligase, was shown to interact with tetherin to promote HIV-1 packaging into CD63^+^ endosomes, with the end result being lysosomal degradation of HIV-1 virions and tetherin in a manner reminiscent of tetherin-dependent autophagic degradation of damaged mitochondria in uninfected cells [190,191].

On the other hand, tetherin has been demonstrated to recruit a second restriction factor, MARCH8, in order to prevent type I IFN signaling in an autophagy-dependent manner [192]. MARCH8 was recently identified as a novel antiretroviral restriction factor in terminally differentiated myeloid cells, such as macrophages and DCs, and is recruited by tetherin to ubiquitinate the RLR MAVS, leading to NDP52-dependent autophagic degradation of MAVS and, as a result, inhibition of MAVS-dependent type I IFN signaling [25,192]. The restrictive action of MARCH8 may also be autophagy-dependent, as MARCH-8-directed degradation of VSV-G is not affected by treatment with a proteasome inhibitor, but was blocked upon treatment with lysosomal protease inhibitors [25]. Next to MARCH8, the closely related MARCH2 protein has a similar mechanism of virus restriction and potential role for autophagy [43]. It would therefore be interesting to investigate if pharmacological autophagy enhancement could promote MARCH8-or MARCH2-mediated HIV-1 restriction.

## 5. TRIM5 as An Innate Immune Sensor

By definition, restriction factors must recognize pathogen-associated molecular patterns (PAMPs) to exert their functions and interfere with the replication cycles of viruses. Recent literature highlights that restriction factors can utilize this function not only to prevent viral replication, but also to act as sensors to initiate innate immune signaling pathways to prevent infection [45,46,47,48,49].

The restriction factor huTRIM5α is upregulated by IFNα signaling, as demonstrated by an increase in TRIM5α mRNA expression levels upon IFN treatment, thereby intensifying blocks in HIV-1 infection [63]. However, huTRIM5α is now understood to act not only as a retroviral restriction factor, but also as a PRR for sensing of incoming retroviral capsids (Figure 3) [49].

When huTRIM5α is overexpressed, or recognizes a retroviral capsid, its RING domain recruits E2 ubiquitin-conjugating enzymes, thereby inducing ubiquitin-dependent signaling [49]. The multimerization of huTRIM5α upon binding the retroviral N-MLV capsid results in a unique trimeric RING domain arrangement that specifically promotes polymerization of K63-linked ubiquitin chains at its N-terminus, as demonstrated in feline (CrFK) and human (TE671) cell lines [80]. This polyubiquitination of huTRIM5α subsequently activates the TAK1 (also known as MAP3K7) kinase complex, and later the AP-1 and NF-κB signaling pathways [49]. This ultimately results in activation of the transcription factors AP-1 and NF-κB, which promote transcription of inflammatory mediators such as CXCL9, CXCL10, CCL8, IL-6, IL-8, and COX2, as shown in primary monocyte-derived DCs and macrophages, THP-1 monocytes and macrophages, and owl monkey kidney (OMK) cells upon recognition of the N-MLV capsid [49]. In addition, OMK cells challenged with VSV-G-pseudotyped HIV-1 upregulated secretion of IL8, CCL2, CCL4, and CSCL10 [49]. Alternatively, in 293T cells, huTRIM5α has also been demonstrated to negatively regulate NF-κB activation, via huTRIM5α-directed degradation of TAK-1 binding protein (TAB)-2, in a dose-dependent manner [193]. The use of proteasomal inhibitors showed that this mechanism is independent of E3 ubiquitin ligase activity, suggesting a role for an alternative degradative pathway. Altogether, this underlines huTRIM5α as a key PRR and regulator of AP-1 and NF-κB and AP-1 and NF-κB-responsive inflammatory mediators, in response to sensing retroviral capsids.

Strikingly, recent data highlight a role for autophagy in huTRIM5α’s functioning as an innate sensor. Genetic depletion of key autophagy proteins such as Beclin 1, ATG7, and ULK1 was required for huTRIM5α-based activation of AP-1 and NF-κB, and thereby of AP-1 and NF-κB-responsive genes [194]. In primary immune cells, induction of IFN-β expression in response to huTRIM5α-restricted recognition of HIV-1 capsid mutant (P90A) was dependent on Beclin 1, ATG7, and ULK1. Furthermore, in autophagy deficient cells, huTRIM5α-driven TAK1 activation, which is also Beclin 1 and ATG7-dependent, was attenuated [194]. Taken together, these data underline that autophagy contributes not only to huTRIM5α-mediated HIV-1 restriction, but also to the role of huTRIM5α as an innate immune sensor of incoming retroviral capsids.

Notably, in primary human DCs, which are a key cell type for linking innate and adaptive immune responses, endogenous huTRIM5α has been demonstrated to negatively regulate induction of type I IFN responses, as well as to positively regulate AP-1 and NF-κB-responsive inflammatory mediators, upon HIV-1 infection [49,62]. Type I IFN suppresses viral replication by upregulating IFN-stimulated genes (ISGs) that can target and inhibit HIV-1 at various steps of the viral life cycle [24,38,144,195]. Activation of IFNα upregulates transcription of ISGs including APOBEC3G, tetherin, and TRIM5α itself, as well as other members of the TRIM family that participate in anti-HIV-1 immune responses such as TRIM22 [44,196,197]. Recent data indicate that upregulation of IFNα is also required for huTRIM5α-mediated HIV-1 restriction in the typically HIV-1 permissive U87-MG.CD4^+^CXCR4^+^ cell line and in primary human CD4^+^ T cells [65]. Altogether, these findings further underline that huTRIM5α plays important intracellular roles in antiretroviral immunity and possesses the capacity for retroviral restriction; however, additional host factors or cytoplasmic counterparts are required to elicit such huTRIM5α-mediated HIV-1 restriction.

In addition to huTRIM5α sensing of exogenous retroviruses such as HIV-1, recent data suggest that the hypothesis that TRIM5α evolution was driven by retroelements, or endogenous retroviruses, is well-founded [186,198,199]. Notably, huTRIM5α was shown to both sense and restrict LINE-1 transposable elements, the only known autonomously active transposable element in humans, in HEK 293T cells [186]. Functional SPRY and B-box domains were essential for huTRIM5α-mediated inhibition of LINE-1 replication and insertion into the host genome. Cytoplasmic co-localization of huTRIM5α and LINE-1 ribonucleoproteins (RNPs), in either cytoplasmic bodies or in a more diffuse pattern, was necessary for huTRIM5α-mediated LINE-1 restriction. In addition, upon sensing LINE-1 RNPs, huTRIM5α activated the TAK1 kinase complex, leading to subsequent activation of the AP-1 and NF-κB pathways [186]. Furthermore, rhTRIMCyp also inhibited LINE-1 retrotransposition, suggesting that, as with the SPRY domain of huTRIM5α, CypA may also bind LINE-1 RNPs leading to retroelement restriction. Altogether, these findings indicate that the intrinsic ability of huTRIM5α to restrict LINE-1 elements is reminiscent of the potency of huTRIM5α to restrict exogenous retroviral capsids.

## 6. *In Vivo* Relevancy of TRIM5α and Autophagy in HIV-1 Infections

### 6.1. Impact of huTRIM5α Polymorphisms on Disease Progression

To further define the roles of huTRIM5α and its relevance as a therapeutic target to treat HIV-1 infection, it is enlightening to examine the presence of single nucleotide polymorphisms (SNPs) in the huTRIM5α gene, and their association with clinical outcomes and disease progression. Several epidemiological genetic studies have demonstrated a relationship between SNPs of huTRIM5α and HIV-1 susceptibility, disease progression and innate and adaptive immunity, as summarized in Table 1.

With regard to HIV-1 susceptibility and disease progression, two SNPs that have been extensively studied are the arginine to glutamine mutation at gene position 136 (R136Q; *TRIM5* rs10838525), and the histidine to tyrosine mutation at position 43 (H43Y; *TRIM5* rs3740996). The former, *TRIM5* R136Q, is located within the CC domain of huTRIM5α and has been demonstrated to have a protective effect on HIV-1 susceptibility in diverse cohorts [200,201,202]. In concordance, *in vitro* studies using Cf2TH cells expressing wild-type (wt) or mutant huTRIM5α and incubated with GFP-reporter HIV-1 have demonstrated that the 136Q variant of huTRIM5α showed superior effects in restricting HIV-1 over wild-type 136R huTRIM5α [200].

Likewise, *TRIM5* H43Y, located in the RING domain, has been shown to have a protective effect on HIV-1 susceptibility, although there are discrepancies in the demonstrated efficacy of this SNP depending on the ethnicity of participants [200,203,204,205]. Thus, it may be that the protective effect of *TRIM5* H43Y on HIV-1 susceptibility depends to some extent on interactions with other polymorphic genes that differ across genetically different populations. Contrastingly, the homozygous *TRIM5* H43Y genotype was found to be predictive for an accelerated disease progression *in vivo*, and deleterious *in vitro* in CRFK cells stably transduced with H43Y huTRIM5α [95,206]. However, this may be explained by the fact that the RING domain is involved not only in ubiquitination of huTRIM5α-HIV-1 complexes but also in self-ubiquitination of huTRIM5α, which is important for homeostatic cycling of the host factor [127]. Thus, in uninfected individuals, mutations such as H43Y in the RING domain of huTRIM5α may result in impaired self-ubiquitination and increased levels of huTRIM5α in the cytoplasm, leading to increased antiretroviral activity against incoming HIV-1 virions and hence an increased protective effect against HIV-1 infection. On the other hand, the E3 ligase activity of the RING domain has also been shown to be involved in the innate immune response against HIV-1 [49]. Therefore, in infected individuals, mutations in the RING domain of huTRIM5α may be associated with decreased control of HIV-1 infection by the innate immune system, thus explaining the association of SNP H43Y with accelerated disease progression.

More recently, several SNPs in huTRIM5α have been demonstrated to be more prevalent in HIV-1 infected patients as compared to healthy controls, indicating a detrimental effect of these polymorphisms on HIV-1 susceptibility. For example, expression of a G249D allele of huTRIM5α SNP rs11038628 was found to be significantly higher in HIV-infected patients as compared to healthy subjects in an Indian population, and to attenuate HIV-1 restricting activity of huTRIM5α *in vitro* [207]. The same tendency, though weaker and non-significant, was also observed in a Japanese population [207].

Similarly, a huTRIM5α variant with the G110R SNP in the B-box domain, which is more frequent in Japanese HIV-1 infected patients than controls, also showed a decreased ability to control HIV-1 *in vitro* [204]. Interestingly, the same study also identified a truncated G176del huTRIM5α variant found only in Japanese populations, which enhanced antiviral activity of co-expressed full-length huTRIM5α. This may be due to the G176del huTRIM5α variant’s enhanced functioning as an activator of innate immune responses, in line with the alternative functioning of huTRIM5α as a PRR as discussed above in Section 5 [204]. Finally, a polymorphic variant of huTRIM5α rs904375, located in intron 4, has been described to affect susceptibility to HIV-1 infection, likely by impacting transcription factor-binding sites and thereby *TRIM5* transcriptional activity [208].

To our knowledge, no studies specifically aiming to determine a correlation between huTRIM5α SNPs and the anti-HIV-1 adaptive immune responses in PLWH have thus far been conducted. However, it has been demonstrated that the synonymous SNPs, *TRIM5* rs3824949 and *TRIM5* rs7122620, correlated with increased levels of rubella- and measles-specific antibodies in humans upon vaccination against these pathogens [209,210]. Notably, patients possessing *TRIM5* rs3824949 who were treated for chronic hepatitis C virus (HCV) were also more likely to reach rapid virologic response (RVR), early virologic response (EVR) and sustained virologic response (SVR) as compared to wt control patients [211]. The beneficial effect of *TRIM5* rs3824949 in achieving SVR to HCV was also observed in HIV/HCV co-infected patients [212]. Moreover, the presence of SNPs *TRIM5* rs7122620 and *TRIM5* rs11820502 correlated, respectively, with increased IL-2 and IL-6 secretion upon measles vaccination [210].

These data illustrate the associations of human *TRIM5* SNPs to differing HIV-1 susceptibility and disease progression underline the *in vivo* relevancy of huTRIM5α. Further underlining the importance of huTRIM5α *in vivo*, higher expression levels of huTRIM5α, as well as TRIM22 and TRIM28, was demonstrated in PLWH who initiated cART treatment early, versus in those who initiated ART treatment during chronic infection phase [213]. However, no relationship between latent HIV-1 reservoir size and huTRIM5α expression in cART-treated patients was demonstrated, although a second study demonstrated increased expression of huTRIM5α, as well as other restriction factors, over time in HIV-1 infected, untreated patients [213,214].

Taken together, these results illustrate that polymorphisms and differential expression of the huTRIM5α gene may impact HIV-1 susceptibility and influence HIV-1 disease progression. Expression of the huTRIM5α gene and its polymorphisms can serve as a predictive biomarker to determine susceptibility to HIV-1 by HIV-1 negative individuals, as well as function as a prognostic biomarker to predict disease progression in PLWH. This utilization of *TRIM5* as a biomarker may assist in the design of optimal treatment regimens and prevention of HIV-1 in the future.

### 6.2. TRIM5α-Based Therapeutic Strategies for HIV-1

In accordance with the long-accepted hypothesis that only rhTRIM5α is able to restrict HIV-1, attempts have been made to genetically engineer rhTRIM5α-expressing human cells. In terms of HIV-1 therapies, the high potential of TRIM-based gene therapy strategies has been extensively reviewed [17]. One potential strategy has been to develop human mimics of TRIMCyp fusion proteins, which restrict HIV-1 when expressed in cell lines or in primary T cells and macrophages [19,85]. Interestingly, a fusion protein of TRIM21 and CypA, TRIM21Cyp, was also effective in preventing HIV-1 reverse transcription in human cell lines and human primary peripheral blood mononuclear cells (PBMCs) [85]. TRIM21Cyp restriction was only partially inhibited by treatment with the protease inhibitor MG132, which may be explained by TRIM21 also being a recruiter of key autophagy machinery upon binding its target proteins, including ULK1, Beclin1, ATG16L1, and GABARAP [85,215]. This could result in autophagic degradation of TRIM21Cyp-HIV-1 complexes. Fusions of CypA with TRIM1, TRIM18, and TRIM19 were also able to restrict HIV-1, with TRIM1Cyp being particularly potent, perhaps because TRIM1 is also an autophagy regulator [129,216].

The potential of mutant huTRIM5α proteins and chimeric rh/huTRIM5α proteins has also been explored. Transduction of CD34^+^ human hematopoietic stem cells (HSCs) with a chimeric rh/huTRIM5α, in combination with a CCR5 short hairpin RNA (shRNA) and a TAR decoy, was shown to be effective in protecting against HIV-1 challenge and providing a selective survival advantage in human CD4^+^ T cells upon infection in a humanized mouse model [217]. A rh/huTRIM5α chimeric variant was also demonstrated to efficiently restrict HIV-1 in HSC-derived macrophages *in vitro* and T-cells *in vivo* in a humanized mouse model transplanted with human fetal tissue [218]. Besides chimeric variants, single amino acid changes at positions 332 and 335 of the huTRIM5α SPRY domain to more closely mimic the rhTRIM5α v1 sequence were also shown to prevent the spread of HIV-1 infection by 20- to 50-fold in TE671 fibroblasts and in the human T-cell line SUP-T1 [97].

However, gene therapy with such variants of TRIM5α has significant downsides. For one, it is possible that huTRIM5α mutants or human mimics of TRIMCyp fusion proteins could disrupt endogenous huTRIM5α function, for example, in innate immune signaling [85]. Furthermore, delivery of these treatment options would entail either transduction and re-infusion of differentiated autologous cells, which would require life-long therapy, or stem-cell based approaches that would necessitate invasive and relatively high-risk preparative ablative chemotherapy [17,219]. Alternatively, targeting TRIM5α-directed cellular mechanisms such as proteasomal activity, autophagy-mediated degradation, and antiviral inflammatory signaling may represent innovative and relevant approaches to capitalize on TRIM5α-mediated antiviral functions.

### 6.3. Impact of Autophagy on HIV-1 Disease Progression and Comorbidities

As outlined above in Section 4, HIV-1 has been demonstrated to evade autophagy in several different cell types *in vitro*, including DCs, macrophages, and CD4^+^ T cells via viral molecules such as Vif and Nef [64,153,184,185]. *In vivo* and *ex vivo* studies have further underlined the importance of autophagy in anti-HIV-1 immunity. *Ex vivo* studies using PBMCs derived from PLWH have shown that long-term non-progressor PLWH contain more specialized autophagy vesicles than those from normal progressors [220]. Notably, viral production and viral load in *ex vivo* PBMCs derived from PLWH were reduced upon treatment with the autophagy-enhancing drug rapamycin [220]. In line with this finding, enhancement of autophagy in monocyte-derived DCs and macrophages and CD4^+^ T cells *in vitro* has shown similar successes in controlling HIV-1 infection or promoting autophagy-dependent apoptosis of HIV-1 infected cells [164,221,222]. Successful control of HIV-1 infection is likely dependent on the level of induced autophagy. Productive HIV-1 infection has been detected in T cell lines and primary human monocyte-derived macrophages with moderate amounts of autophagosomes present, whereas strong autophagy induction inhibits viral replication and infection [223]. This indicates that intrinsic autophagy mechanisms, and, more importantly, pharmaceutical enhancement of autophagy, can control HIV-1 infection.

Furthermore, autophagy dysfunction has been associated with increased risk for HIV-1 associated comorbidities in cART-treated HIV-1 patients and animal models [224,225]. Cardiovascular disease is a leading comorbidity for PLWH, even in those who are cART-treated [226]. The HIV-1 Nef protein, which is thought to be released extracellularly by HIV-1 infected CD4^+^ T cells and is detectable in the serum of HIV-1-positive patients, accumulates in cardiomyocytes where it dysregulates autophagy leading to cardiomyocyte cytotoxicity and death. Strikingly, this effect could be reversed *in vitro* by treatment with the autophagy-inducing drug rapamycin [225]. Nef is not the only HIV-1 protein affecting cardiomyocytes; the HIV-1 protein Tat also has a profound impact on these cells. Recent research has shown that transduction of cardiomyocytes with adenoviral-Tat impaired mitochondrial Ca2^+^ uptake and the electrophysiological activity of the cardiomyocytes. In addition, Tat impaired the protein clearance function of autophagy under hypoxic conditions, due to a reduction in ubiquitination and dysregulation of autophagy proteins p62 and LC3-ll [227]. As well as Nef and Tat, the HIV-1 proteins Env, Gag, Vif, and p24 have all been shown to impact autophagy pathways or serve as specific substrates for autophagy, depending on the cell type, and thereby evade and dysregulate the autophagy machinery as reviewed in [228].

The central nervous system is also an important target of HIV-1, and neurological impairment ranging from severe HIV-1 associated dementia in untreated PLWH to minor cognitive disorders in cART-treated PLWH have been described [229,230]. In post-mortem brains from HIV-1 patients with encephalitis, levels of autophagy proteins such as Beclin 1, ATG5, ATG7, and LC3-II were significantly increased, as compared to the brains from HIV-1 positive patients without HIV-1 encephalitis or uninfected controls [224]. In line with this finding, *in vitro* studies showed that adding the supernatant from HIV-1 infected primary macrophages to human fetal neurons resulted in the increase in autophagy-proteins Beclin-1 and ATG5, but no LC3-ll formation was detected [231]. This suggests that while the early steps of autophagy were promoted in neurons, HIV-1 inhibits the later degradative steps of autophagy leading to protein accumulation and thereby neuronal toxicity. Strikingly, the same study showed that autophagy-inducing rapamycin treatment of neurons resulted in protection against HIV-1 toxicity [231]. Taken together, these data are illustrative of widespread autophagy dysregulation during HIV-1 disease progression. Nevertheless, the *in vivo* relevancy of autophagy in non-progressor PLWH and the beneficial effects of autophagy inducers *in vitro* showcase the protective role of autophagy mechanisms in HIV-1 disease progression and comorbidity development, and the therapeutic potential to harness autophagy for HIV-1 management.

### 6.4. Harnessing Autophagy for HIV-1 Management

As demonstrated by several *in vitro* and *ex vivo* studies, boosting autophagy in HIV-1 target cells may be an economically feasible and therapeutically effective way to treat HIV-1 infection. Indeed, autophagy-targeting drugs have already proved effective in intervening in HIV-1 infection in human macrophages, DCs, PBMCs, and CD4^+^ T cells [164,220,221,222,232]. For example, TLR8-induced autophagy inhibits HIV-1 release by infected human monocyte-derived macrophages via a vitamin-D dependent mechanism, and treatment of human monocyte-derived macrophages with vitamin D promoted autophagy-mediated inhibition of HIV-1 replication [221,233]. Treatment with the autophagy-enhancing peptide Tat-vFLIP-α2 could also initiate autophagy-dependent cell death in HIV-1 infected human macrophages [232]. Similarly, treatment with a Tat-Beclin 1 peptide also induced autophagy and had a protective effect against HIV-1 infection of human monocyte-derived macrophages [234]. Comparable effects have been demonstrated in *ex vivo* PBMCs and CD4^+^ T cells derived from HIV-infected individuals, where treatment with the autophagy enhancing drug rapamycin or DIABLO/SMAC mimetics, respectively, reduced HIV-1 replication, or selectively induced killing of latently infected CD4^+^ T cells [220,222]. Likewise, rapamycin treatment could also reverse the HIV-1 Nef-induced block in autophagosome maturation in cardiomyocytes [225].

Strikingly, we have recently demonstrated that treatment with autophagy-enhancing drugs limits HIV-1 acquisition and suppresses ongoing HIV-1 replication in *ex vivo* human tissue-derived CD11c^+^ DCs and CD4^+^ T cells [164]. These data were also corroborated in vaginal and intestinal tissues. Prophylactic treatment with autophagy-enhancing drugs carbamazepine or everolimus significantly reduced HIV-1 acquisition by subepithelial CD11c^+^ DCs and CD4^+^ T cells. Notably, a cell-specific effect of post-infection treatment with autophagy drugs was observed. Treatment with carbamazepine, everolimus, or rapamycin reduced HIV-1 replication in infected *ex vivo* CD4^+^ T cells, while only therapeutic rapamycin administration intervened in ongoing HIV-1 infection in CD11c^+^ DCs. Taken together, these data support harnessing autophagy as a relevant emerging target for HIV-1 therapies [164].

Such autophagy-enhancing drugs may circumvent HIV-1-mediated down-regulation of the autophagy system in distinct HIV-1 target cells without usurping alternative signaling roles of the endogenous proteins. This could permit not only more efficient huTRIM5α-mediated HIV-1 restriction in immune cells such as LCs, which possess an intrinsic capacity for such restriction, but also promote huTRIM5α-mediated HIV-1 restriction in permissive immune cells such as DCs and CD4^+^ T cells, in which HIV-1 likely circumvents huTRIM5α-mediated restriction by blocking autophagy. Finally, autophagic degradation of non-self peptides is well-established to promote antigen presentation [235,236]. Thus, degradation of HIV-1 via autophagy could potentially promote presentation of HIV-1 specific antigens by DCs, enhancing the HIV-1 specific T-cell adaptive immune response. In summary, the use of autophagy enhancers in combination with current cART therapies may have the potential to both reboot the restrictive capacities of host restriction factors for HIV-1 in primary human immune cells and maximize the antiviral immune response mediated by huTRIM5α, as well as other autophagy-interacting restriction factors.

## 7. Conclusions and Outlook

To reiterate previous reviews, a silver-bullet strategy for HIV-1 cure is unlikely to work, given the hypervariability of the retrovirus and its many adaptations to evade host immune responses and establish viral reservoirs [17,237]. Rather, we should hedge our bets on exploiting a panel of strategies to eradicate HIV-1, including leveraging natural restriction factors, such as TRIM5α, and harnessing the power of protective host pathways, such as autophagy and the immunoproteasome, to prevent infection and promote innate immune responses. Here, we have provided an overview of the multifaceted functions of TRIM5α, which span from virus sensing and restriction to induction of innate immune responses. We have also outlined the impact of TRIM5α genetic polymorphisms on HIV-1 susceptibility and HIV-1 disease progression and comorbidities of PLWH. We propose a novel conceptualization of TRIM5α-mediated HIV-1 restriction. Until recently, TRIM5α-mediated HIV-1 restriction has primarily been considered to be limited to the rhesus orthologue, which relies on early disintegration of the HIV-1 capsid and subsequently proteasomal degradation to exert its protective function. The demonstrated functionalities of huTRIM5α to prevent LINE-1 retrotransposition, restrict HIV-1 in primary human LCs and in DCs treated with SUMOylation inhibitor, and mediate immunoproteasome-dependent restriction of HIV-1 in CD4^+^ T cells underpin the intrinsic ability of the human huTRIM5α orthologue to restrict HIV-1 infection [62,64,65,186].

Based on this recent literature, we propose that TRIM5α should be conceptualized as a cell-specific restriction factor with a varied portfolio of protective functions, including functionality as an autophagic effector, innate immune retroviral sensor, promoter of viral degradation, and potentially a linker of innate antiviral and adaptive immunity. We have also expanded on the roles of autophagy in the context of HIV-1, and propose that pharmacological modulation of autophagy mechanisms may be a financially and therapeutically efficacious treatment option in combination with current ART.

## Figures and Tables

**Figure 1 viruses-13-00320-f001:**
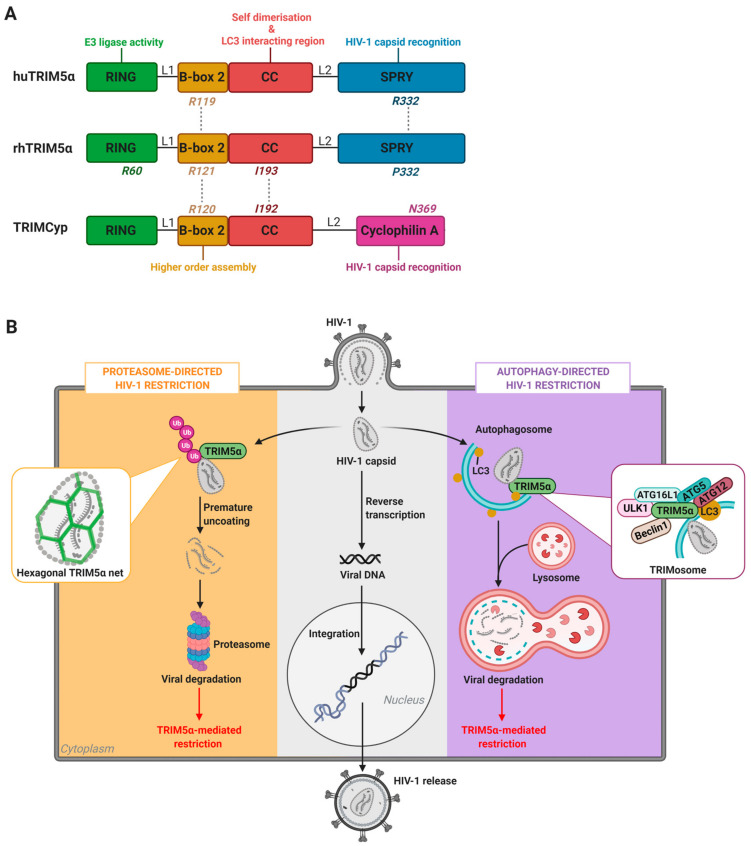
Structure and functions of TRIM5α. (**A**) Rhesus (rh)TRIM5α, human (hu)TRIM5α, and TRIMCyp protein domain organization and associated functions. Key amino acid residues impacting antiviral functions per protein domain are depicted. (**B**) Schematic overview of the main steps of HIV-1 virus replication cycle (center panel, in grey), and the proteasome-directed (left panel, in orange) and autophagy-directed (right panel, in purple) TRIM5α-mediated mechanisms of HIV-1 restriction. RING: really interesting new gene, CC: coiled-coil, LC3: microtubule-associated protein 1A/1B-light chain 3, ULK1: Unc-51 like autophagy activating kinase 1, ATG: autophagy related, Ub: ubiquitin, L1: linker 1, L2: linker 2.

**Figure 2 viruses-13-00320-f002:**
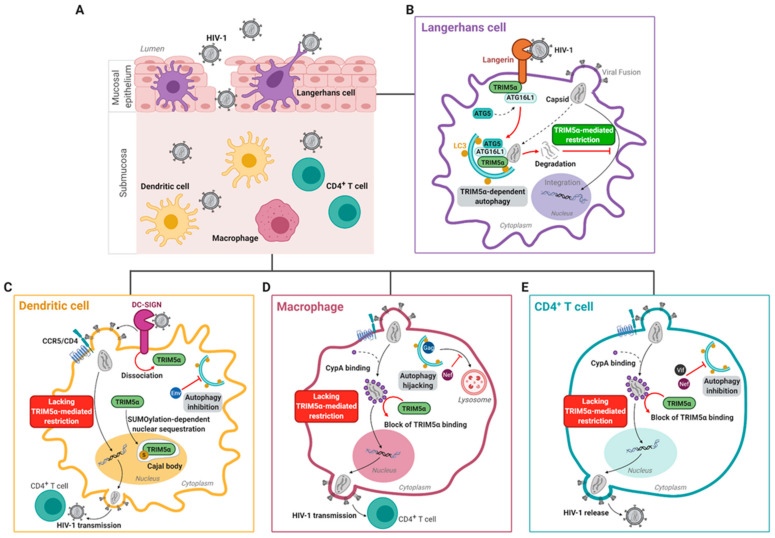
Cell-specific TRIM5α-mediated HIV-1 restriction. Langerhans cells, which reside within the mucosal epithelium (**A**), are able to restrict HIV-1 infection via a TRIM5α-mediated mechanism (**B**). At steady-state, the Langerhans cell-specific receptor Langerin is in association with a TRIM5α-ATG16L1 complex. Binding of Langerin to HIV-1 results in the internalization of the virus. Upon viral fusion, human TRIM5α mediates recruitment of ATG5 to form the TRIM5α–ATG16L1–ATG5–HIV-1p24 capsid complex, which promotes autophagosome formation and subsequent degradation of the viral capsid, thereby preventing HIV-1 infection and transmission by human Langerhans cells. Contrastingly, human and non-human primate dendritic cells, which reside within the submucosa (**A**), lack efficient TRIM5α-mediated retroviral restriction (**C**). Submucosal dendritic cells capture HIV-1 via the receptor DC-SIGN, which promotes productive HIV-1 infection of dendritic cells. Upon binding of HIV-1 to dendritic cells, TRIM5α dissociates from the DC-SIGN signalosome, and TRIM5α is trapped in nuclear Cajal bodies via a SUMOylation-dependent manner, thereby preventing cytosolic TRIM5α from forming a complex with the incoming viral capsid. In addition, autophagy is inhibited by the HIV-1 protein Env. Altogether, this results in HIV-1 escape from human TRIM5α/autophagy-mediated HIV-1 restriction mechanisms. (**D**) Upon fusion of the viral capsid with the macrophage membrane, the host protein CypA readily binds the incoming HIV-1 capsid. CypA coating of the capsid blocks the interaction of human TRIM5α with the viral capsid, and thereby prevents human TRIM5α-mediated restriction in macrophages. In macrophages, HIV-1 hijacks the early stages of the autophagy pathway for HIV-1 Gag processing. In addition, HIV-1 Nef inhibits the late degradative stages of autophagy thereby preventing lysosomal viral degradation. (**E**) The autophagy machinery is also downregulated in CD4^+^ T cells by the HIV-1 proteins Vif and Nef. HIV-1 can thereby integrate into the host genome in CD4^+^ T cells unhindered, to replicate and be released. Similar to macrophages, upon fusion of the viral capsid with the CD4^+^ T cell membrane, CypA binds the incoming HIV-1 capsid, preventing human TRIM5α-mediated restriction in CD4^+^T cells. S: small ubiquitin-like modifier (SUMO), CypA: Cyclophilin A.

**Figure 3 viruses-13-00320-f003:**
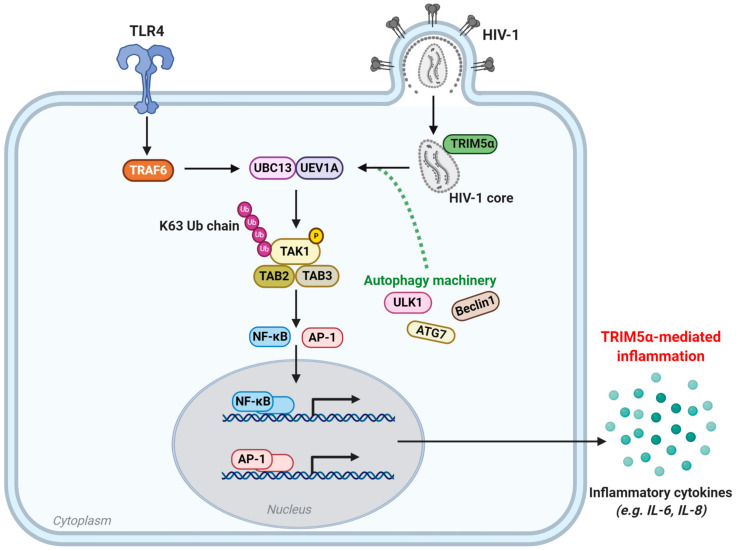
TRIM5α-mediated regulation of innate immune signaling and molecular inflammation. The TAK1 signaling pathway is activated upon TLR4 stimulation, or upon sensing of viral capsid by TRIM5α. Subsequently, the E3 ligase activity of TRAF6 is activated and, together with the E2 ubiquitin conjugating enzyme complex UEV1A-UBC13, promotes K63-chain linked ubiquitination and autophosphorylation of the TAK1-TAB complex. In the context of HIV-1 infection, sensing of the viral capsid by TRIM5α initiates its E3 ligase activity resulting in K63-chain linked ubiquitination and autophosphorylation of the TAK1-TAB complex. Subsequently, the TAK1-TAB complex activates the transcription factors NF-kB and AP-1, leading to the transcription and ultimately secretion of pro-inflammatory cytokines such as IL-6 and IL-8. The autophagy-related proteins ULK1, Beclin1 and ATG7 are essential for TRIM5α to function as an innate immune sensor. However, precisely how they contribute to the TRIM5α-mediated inflammation is as yet uncertain. TLR4: toll-like receptor 4, TAK1: transforming growth factor-β-activated kinase 1, TAB: TAK1-binding protein, UEV1A: ubiquitin-conjugating enzyme E2 variant 1A, UBC13: ubiquitin-conjugating enzyme E2 13, AP-1: activating protein 1, NF-κB: nuclear factor κB, Ub: ubiquitin, P: phosphate, IL: interleukin.

**Table 1 viruses-13-00320-t001:** Association between hu*TRIM5* SNPs and HIV-1 susceptibility, disease progression, and immunity.

Effect	SNP	Domain	Implication	Outcome	*p*	Ref.
HIV-1 susceptibility	rs3740996	RING	Higher frequencies in SN ^1^ compared to SC ^2^	Protective	<0.043	[200,205]
			No association with HIV-1 susceptibility	No effect	-	[203]
	rs10838525	CC	Higher frequencies in HREU ^3^ compared to SC	Protective	<0.02	[200,201]
	rs904375	Intron 4	Higher frequencies in HIV-1 positive individuals compared to healthy controls	Negative	0.46	[208]
	rs11038628	Linker 2	Higher frequencies in HIV-1 positive individuals compared to healthy controls	Negative	0.026	[207]
	G110R	B-box	Higher frequencies in HIV-1 positive individuals compared to healthy controls	Negative	0.002	[204]
Disease progression	rs3740996	RING	Homozygous genotype resulted in accelerated disease progression	Negative	-	[95,206]
	rs10838525	CC	Protective effect observed only when CXCR4-using HIV-1 variants were present and when a viral load of 10 ^4,5^ copies/mL plasma was used as endpoint	Partial effect	0.008 (HZ ^4^); 0.03 (HO ^5^)	[206]
	rs3824949	5′ UTR	Accelerated disease progression when in combination with the 136Q allele (rs10838525)	Negative	0.009	[206]
Innate; adaptive immunity	rs3824949	5′ UTR	Improved RVR ^6^, EVR ^7^, SVR ^8^ for Hepatitis C virus	Protective	<0.001	[211]
	rs3824949 rs7122620	5′ UTR 3′ UTR	Allele dose-related increase in rubella antibody response	Protective	<0.015	[209]
			Differences in measles specific antibody levels.	Protective	<0.02	[210]
	rs7122620	3′ UTR	Increased IL-2 secretion levels upon measles vaccination	Protective	<0.015	[210]

^1^ SN: HIV-1 seronegative individual, belonging to an HIV-1 risk group; ^2^ SC: HIV-1^+^ seroconverter (weeks after HIV-1 infection during first antibodies against HIV-1 appear); ^3^ HREU: high-risk exposed HIV-1 uninfected. ^4^ Heterozygous for the SNP; ^5^ homozygous for the SNP; ^6^ RVR: rapid virologic response; ^7^ EVR: early virologic response; ^8^ SVR: sustained virologic response; UTR: untranslated region.

## Data Availability

No new data were created or analyzed in this study. Data sharing is not applicable to this article.

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
