# Peer review of "Human TRIM5α: Autophagy Connects Cell-Intrinsic HIV-1 Restriction and Innate Immune Sensor Functioning"

_viruses, 2021, doi:10.3390/v13020320_

Round 1
Reviewer 1 Report
The review article by Cloherty et al provides an overview of the antiviral host restriction factor TRIM5a in the context of HIV infection. The composition of the article is such that the reader can find all relevant information within the text. The article frequently compares the highly efficient macaque TRIM5a and the less efficient human TRIM5a, which is a good approach to highlight the differential activity of these molecules. The article is very well written and presented.
Author Response
Response to reviewer 1
The review article by Cloherty et al provides an overview of the antiviral host restriction factor TRIM5a in the context of HIV infection. The composition of the article is such that the reader can find all relevant information within the text. The article frequently compares the highly efficient macaque TRIM5a and the less efficient human TRIM5a, which is a good approach to highlight the differential activity of these molecules. The article is very well written and presented.
- We thank the reviewer for the positive feedback.

Reviewer 2 Report
The authors described the molecular function of TRIM5α function in HIV-1 lifecycle in this manuscript. The authors clarified the importance and regulatory mechanisms of TRIM5α in HIV-1 infection. Overall, the manuscript looks very interesting and well described in a logical order. However, there are several minor points that need to be addressed.
- It would be highly appreciated if the authors could give a brief introduction on TRIM5α discovery in section 1.
- In figure 1A, it would be much more informative if the authors could include the structures of TRIM5α from different primate species, and highlight the essential domains and key amino acids determining its antiviral activity.
- I felt that perhaps at some places there is some repetition, or some of the more general information interspersed throughout could be mention early on in the introduction. but this is not a big issue at all, and can be looked at during a later editing stage.
Author Response
Response to reviewer 2
The authors described the molecular function of TRIM5α function in HIV-1 lifecycle in this manuscript. The authors clarified the importance and regulatory mechanisms of TRIM5α in HIV-1 infection. Overall, the manuscript looks very interesting and well described in a logical order. However, there are several minor points that need to be addressed.
Minor points:
- Point 1: It would be highly appreciated if the authors could give a brief introduction on TRIM5α discovery in section 1.
- Response 1: We agree with the reviewer. A brief introduction to the discovery of TRIM5α has been added to section 1.2. We have provided more detail regarding the experiments that uncovered the protective effect of rhesus macaque TRIM5α and situated these ground breaking findings within the framework of the previously demonstrated species-specific block of HIV-1 prior to reverse transcription that was observed in Old World Monkeys, but not in humans.
Below the exact textual changes as it appears in the revised version of the manuscript:
Section 1.2 (lines 133-156): "Already in the late 1990s, data emerged to demonstrate a species-specific block of HIV-1 prior to reverse transcription occurring in Old World Monkeys, but not humans [51–55]. However the responsible species-specific dominant repressive factor targeting the incoming HIV-1 capsid remained undiscovered until 2004. Stremlau and colleagues identified TRIM5α as the post-entry host factor potently restricting HIV-1 [50]. Expression of rhesus macaque TRIM5α (rhTRIM5α) in HeLa cells led to significantly decreased HIV-1 infection as compared to HeLa cells containing empty vectors. However, although expression of rhTRIM5α in HeLa cells efficiently blocked HIV-1 infection, Simian Immunodeficiency Virus (SIV) was less restricted [50]. Furthermore, ectopic expression of the human TRIM5α orthologue (huTRIM5α) failed to restrict HIV-1 as potently as rhTRIM5α. Taken together, these data led to the conceptualization of TRIM5α as a species-specific HIV-1 restriction factor, with rhTRIM5α but not huTRIM5α efficiently restricting the virus [50]. In the following years, a wealth of literature has since focused on describing this TRIM5α-mediated mechanism of restriction [49,56–63].
It has since been demonstrated that rhTRIM5α restricts HIV-1 infection by forming hexagonal nets around incoming virus capsids and subsequently directing these TRIM5α-HIV-1 complexes for degradation [56–60]. More recently, rhTRIM5α was also reported to act as a pathogen recognition receptor (PRR) by sensing incoming retroviral capsids, and upregulating IFN responses as well as AP-1 and NF-κB-responsive inflammatory mediators [49,61–63]. Notably, recent literature indicates that the huTRIM5α is also a HIV-1 restriction factor, however it functions via a distinct mechanism as compared to rhTRIM5α, and it likely operates in a cell-specific manner [62,64,65]. huTRIM5α antiretroviral activity is also regulated by supplementary host proteins, such as cell-specific receptors and chaperones [64,66]."
- Point 2: In figure 1A, it would be much more informative if the authors could include the structures of TRIM5α from different primate species, and highlight the essential domains and key amino acids determining its antiviral activity.
- Response 2: We thank the reviewer for this suggestion. We have expanded on Figure 1A, and have now included graphical representations from different primate species (huTRIM5α, rhTRIM5α, and TRIMCyp). As suggested, key amino acid residues reported to impacting antiviral TRIM5α activity are now detailed per domain and for each orthogue: namely R60 in the RING domain of rhTRIM5α; R119/121/120 in the B-Box 2 domain of the different orthologues; I193/2 in CC domain of rhTRIM5α and TRIMCyp; R332/P332 in SPRY domain of huTRIM5α and rhTRIM5α; and the TRIMCyp N369 within the cyclophilin A domain of TRIMCyp.
These graphical changes (Figure 1A) have also been reflected in the figure legend and in updated text in the corresponding sections 2.1, 2.2, and 2.3.
Below the exact textual changes as it appears in the revised version of the manuscript:
Figure 1 (legend, lines 208-210): “(A) Rhesus (rh)TRIM5α, human (hu)TRIM5α, and TRIMCyp protein domain organization and their associated functions. Key amino acid residues impacting antiviral functions per protein domain are depicted”
Section 2.1 (lines 181-189): “In vitro mutations of the B-box residue R121 inhibit the formation of hexagonal structures on the surface of HIV-1 capsid, and thereby abrogated rhTRIM5α-mediated HIV-1 restriction (Figure 1, panel A) [58,59,70]. In addition, an I193A mutation within the CC-domain of a rhTRIM5α fusion protein resulted in loss of restriction in HeLa cells and a slight instability of the rhTRIM5α dimer [71]. Structural modeling indicated that residue I193 is likely important for the correct packaging of the CC/L2/SPRY domains, and that I193A mutation may alter the positioning of the SPRY domain relative to the CC-domain, associated with defective binding to the viral capsid [71].”
Section 2.1 (lines 199-206) “The E3 ubiquitin ligase activity of the RING domain of rhTRIM5α is key in directing rhTRIM5α-mediated degradation of the HIV-1 capsid. Demonstrative of this, the R60A mutation within the rhTRIM5α RING domain, which abolishes its E3 ubiquitin ligase activity, interferes with retroviral restriction (Figure 1, panel A) [76,77]. The rhTRIM5α RING domain normally directs the elongation of N-terminally anchored K63-linked ubiquitin chains to the viral capsid, which tags the incoming virus for destruction. These ubiquitin-tagged rhTRIM5α-HIV-1 complexes are then directed to the proteasome for subsequent degradation [56,78–80]. ”
Section 2.1 (lines 218-222): "Proteasome inhibition or introduction of a RING domain mutations C15A or C18A in rhTRIM5α alters the intracellular localization of rhTRIM5α, causing it to accumulate in relatively large cytoplasmic or (peri)nuclear bodies respectively and resulting in decreased availability of rhTRIM5α to restrict HIV-1 [78,81]”
Section 2.2 (lines 246-258): "Recent studies have highlighted that, similarly to rhTRIM5α, the TRIMCyp CC and B-box domains facilitate its dimerization and subsequent multimerization, thereby promoting aggregation into cytoplasmic bodies and, upon HIV-1 infection, formation of hexagonal TRIMCyp nets around target virus capsid proteins [60,91]. Higher-order assembly of TRIMCyp may serve an analogous function of coupling capsid binding and ubiquitination to promote HIV-1 degradation, as with rhTRIM5α.
Like rhTRIM5α, mutating amino acid residue R120 within the TRIMCyp B-box domain abrogates HIV-1 restriction by TRIMCyp (Figure 1, panel A) [70]. In contrast to the rhesus orthologue, mutation of the TRIMCyp CC-domain residue I192 did not result in the abrogation of HIV-1 restriction. [71].
In regards to capsid binding, N369 within the CypA domain of TRIMCyp is a key residue for HIV-1 capsid recognition (Figure 1, panel A)[92]”
Section 2.3 (lines 285-289): “Furthermore, it has also been demonstrated that the huTRIM5α B-box residue R119, like its orthologues rhTRIM5α (R121) and TRIMCyp (R120), is essential for hexagonal structure formation and viral restriction [70]. huTRIM5α R119E and R119D mutants lost antiretroviral activity against non-human viruses such as N-MLV [70].”
- Point 3: I felt that perhaps at some places there is some repetition, or some of the more general information interspersed throughout could be mention early on in the introduction. but this is not a big issue at all, and can be looked at during a later editing stage
- Response 3: We apologize to the reviewer for some repetition. In order to keep the manuscript readable for newcomers to the field, and to tie up the different segments of the manuscript, we have currently opted to incorporate relevant general information at the beginning of each section. If the reviewer wishes, we would be happy to adjust the text accordingly where deemed necessary.

Reviewer 3 Report
The authors provided a detailed overview of TRIM5 alpha function in HIV sensing and restriction, differences in viral restriction across species, impact of TRIM5α genetic polymorphisms on HIV-1 susceptibility and HIV-1 disease progression of infected patients, viral restriction factors and relation to host degradation pathways. TRIM5a viral restriction can be via proteasome-directed and autophagy-directed pathways. Also TRIM5a function in the induction of innate immune responses was discussed. TRIM5a restriction in various cell types such as langerhans cells, dendritic cells and T cells was discussed. The idea is to demonstrate that adjunctive therapies harnessing restriction factors like TRIM5a and protective host pathways like autophagy and innate immune responses would be helpful in the prevention of HIV infection as well as strategies for HIV eradication. It is a very nicely written review covering the TRIM5a functions in detail. However, there are few shortfalls noticed that should be included in the revised manuscript.
The major concern is that after detailed description of how TRIM3a plays an important role with its multifaceted functions in viral restriction and innate immune responses, finally the therapeutic approaches presented were only to target autophagy for HIV degradation. Leveraging TRIM5a dependent approaches, whether they are feasible or not, in enhancing autophagy or innate immune response are not discussed.
Another concern is that discussion on TRIM5a viral restriction in macrophages, which is also a HIV reservoir, is not included.
Author Response
Response to reviewer 3
The authors provided a detailed overview of TRIM5 alpha function in HIV sensing and restriction, differences in viral restriction across species, impact of TRIM5α genetic polymorphisms on HIV-1 susceptibility and HIV-1 disease progression of infected patients, viral restriction factors and relation to host degradation pathways. TRIM5a viral restriction can be via proteasome-directed and autophagy-directed pathways. Also TRIM5a function in the induction of innate immune responses was discussed. TRIM5a restriction in various cell types such as langerhans cells, dendritic cells and T cells was discussed. The idea is to demonstrate that adjunctive therapies harnessing restriction factors like TRIM5a and protective host pathways like autophagy and innate immune responses would be helpful in the prevention of HIV infection as well as strategies for HIV eradication. It is a very nicely written review covering the TRIM5a functions in detail. However, there are few shortfalls noticed that should be included in the revised manuscript.
Point 1: The major concern is that after detailed description of how TRIM5a plays an important role with its multifaceted functions in viral restriction and innate immune responses, finally the therapeutic approaches presented were only to target autophagy for HIV degradation. Leveraging TRIM5a dependent approaches, whether they are feasible or not, in enhancing autophagy or innate immune response are not discussed.
- Response 1: As suggested by the reviewer, we have added an additional section (section 6.2), detailing research regarding TRIM5α-based treatment strategies for HIV-1. We have presented the latest research on developing human mimics of TRIMCyp fusion proteins, mutant huTRIM5α proteins, and chimeric hu/rhTRIM5α proteins. Additionally, we have discussed the therapeutic potential of TRIM-based gene therapy strategies and connected it with autophagy.
Below the exact textual changes as it appears in the revised version of the manuscript:
Section 6.2 (lines 876-912): “6.2. TRIM5α-based therapeutic strategies for HIV-1
In accordance with the long accepted hypothesis that only rhTRIM5α is able to restrict HIV-1, attempts have been made to genetically engineer rhTRIM5α-expressing human cells. In terms of HIV-1 therapies, the high potential of TRIM-based gene therapy strategies has been extensively reviewed [17]. One potential strategy has been to develop human mimics of TRIMCyp fusion proteins, which restrict HIV-1 when expressed in cell lines or in primary T cells and macrophages [19,85]. Interestingly, a fusion protein of TRIM21 and CypA, TRIM21Cyp, was also effective in preventing HIV-1 reverse transcription in human cell lines and human primary PBMCs [85]. TRIM21Cyp restriction was only partially inhibited by treatment with the protease inhibitor MG132, which may be explained by TRIM21 also being a recruiter of key autophagy machinery upon binding its target proteins, including ULK1, Beclin1, ATG16L1, and GABARAP [85,215]. This could result in autophagic degradation of TRIM21Cyp-HIV-1 complexes. Fusions of CypA with TRIM1, TRIM18, and TRIM19 were also able to restrict HIV-1, with TRIM1Cyp being particularly potent, perhaps because TRIM1 is also an autophagy regulator [129,216].
The potential of mutant huTRIM5α proteins and chimeric rh/huTRIM5α proteins has also been explored. Transduction of CD34+ human hematopoietic stem cells (HSCs) with a chimeric rh/huTRIM5α, in combination with a CCR5 short hairpin RNA (shRNA) and a TAR decoy, was shown to be effective in protecting against HIV-1 challenge and providing a selective survival advantage in human CD4+ T cells upon infection in a humanized mouse model [217]. A rh/huTRIM5α chimeric variant was also demonstrated to efficiently restrict HIV-1 in HSC-derived macrophages in vitro and T-cells in vivo in a humanized mouse model transplanted with human fetal tissue [218]. Besides chimeric variants, single amino acids changes at position 332 and 335 of the huTRIM5α SPRY domain, to more closely mimic the rhTRIM5α v1 sequence, were also shown to prevent the spread of HIV-1 infection by 20 to 50 fold in TE671 fibroblasts and in the human T-cell line SUP-T1 [97].
However, gene therapy with such variants of TRIM5α has significant downsides. For one, it is possible that huTRIM5α mutants or human mimics of TRIMCyp fusion proteins could disrupt endogenous huTRIM5α function, for example in innate immune signaling [85]. Furthermore, delivery of these treatment options would entail either transduction and re-infusion of differentiated autologous cells, which would require life-long therapy, or stem-cell based approaches that would necessitate invasive and relatively high-risk preparative ablative chemotherapy [17,219]. Alternatively, targeting TRIM5α-directed cellular mechanisms like proteasomal activity, autophagy-mediated degradation, and antiviral inflammatory signaling may represent innovative and relevant approaches to capitalize on TRIM5α-mediated antiviral functions”
Point 2: Another concern is that discussion on TRIM5a viral restriction in macrophages, which is also a HIV reservoir, is not included.
- Response 2: We thank the reviewer for this suggestion, and agree that a discussion of TRIM5α-mediated viral restriction in macrophages is an important addition to this review. We have now included a section (section 4.2), detailing the role of both TRIM5α and autophagy pathways in HIV-1 restriction in macrophages. We have outlined the interactions between viral protein Nef and host autophagy regulators Beclin1 and TFEB, discussed the capacity of HIV-1 in hijacking early stage of autophagy pathway and the role of host CypA in blocking huTRIM5α-mediated HIV-1 restriction in human macrophages. In addition, we have expanded on Figure 2, and have now included a new graphical representation of the impact of huTRIM5α and autophagy pathways on HIV-1 infection of macrophages (Figure 2D).
Below the exact textual changes as it appears in the revised version of the manuscript:
Figure 2D legend (lines 504-514): “(D) Upon fusion of the viral capsid with the macrophage membrane, the host protein CypA readily binds the incoming HIV-1 capsid. CypA coating of the capsid blocks the interaction of human TRIM5α with the viral capsid, and thereby prevents human TRIM5α-mediated restriction in macrophages. HIV-1 hijacks the early stages of the autophagy pathway for Gag processing. In addition, Nef inhibits the late degradative stages of autophagy thereby preventing lysosomal viral degradation. (E) The autophagy machinery is also downregulated in CD4+ T cells by the HIV-1 proteins Vif and Nef. HIV-1 can thereby integrate into the host genome in CD4+ T cells unhindered, to replicate and be released. S: small ubiquitin like modifier. Similarly to macrophages, upon fusion of the viral capsid with the CD4+ T cell membrane, CypA binds the incoming HIV-1 capsid, preventing human TRIM5α-mediated restriction in CD4+T cells.”
Section 4.3 (583-621): “4.2. Macrophages
Macrophages are myeloid lineage cells that are resident in subepithelial tissues throughout the body. They can originate from self-renewing tissue-resident macrophages or infiltrating blood-derived monocytes [170,171]. As antigen-presenting cells, macrophages are important for bridging innate and adaptive immunity by phagocytosing extracellular pathogens, and presenting pathogen-derived antigens to CD4+ T cells [172]. Because macrophages express the HIV-1 entry receptor CD4 and co-receptors CCR5 and CXCR4 they are themselves susceptible to HIV-1 infection [13]. In addition, HIV-1 infected macrophages can transmit virus to CD4+ T cells via a viral synapse, i.e. cell-to-cell contact, in a manner similar to that of DCs [173–175].
In macrophages, HIV-1 hijacks the early non-degradative stages of autophagy. Research using primary human macrophages, as well as THP-1 and U937 monocyte-derived macrophages, has demonstrated that the induction of autophagy with rapamycin increased extracellular HIV-1 yields [176]. In addition, inhibition of autophagy via treatment with 3MA or knock-out of Beclin1 and ATG7 resulted in a reduction in the extracellular release of infectious virus as compared to untreated cells, indicating that autophagy promotes HIV-1 maturation or release in macrophages [176]. This is in contrast with other immune cell types, as induction of autophagy in DCs and CD4+ T cells results in a decrease in production of mature virions [164,177].
In macrophages, the interaction of toll-like receptor 8 (TLR8) and HIV-1 induces autophagy by Beclin 1-dependent dephosphorylation of transcription factor EB (TFEB) and subsequently its nuclear translocation during the initial stages of HIV-1 infection [178,179]. Upon the induction of autophagy, HIV-1 Gag proteins co-localize with autophagy-related protein LC3 and subsequently promote Gag processing [176].
The HIV-1 Nef protein plays an important role in manipulation of host autophagy, and thereby immune evasion, in macrophages. Several studies have indicated that, by inhibiting Beclin 1 association with class III PI 3-kinase (PI3KC3) complex II, which plays a key role in autophagosome-lysosome fusion, HIV-1 Nef blocks autophagosome maturation [176,180]. Interestingly, recent studies have also indicated that Nef additionally blocks autophagy at its early initiation stages [176,178,180,181].
In addition to evading host antiviral mechanisms in macrophages, HIV-1 also exploits host proteins to evade restriction. Recent studies have demonstrated that multimeric binding of host CypA proteins to the viral capsid prevents interaction of huTRIM5α with the viral core, thereby blocking huTRIM5α-mediated restriction [165,182]. Correspondingly, HIV-1 viruses bearing capsid mutation P90A, which prevents CypA from interacting with the capsid, were restricted by huTRIM5α in macrophages, as was also demonstrated in DCs [66,165–167]. Likewise, a construct generated by fusion of CypA to huTRIM5α (hT5Cyp), inspired by the new world monkey TRIMCyp as described in section 2.2, was able to block HIV-1 infection in human primary macrophages [19]. ”
